# Energetic and physical limitations on the breaching performance of large whales

**Paolo S Segre**[1]\*, **Jean Potvin**[2], **David E Cade**[1], **John Calambokidis**[3], **Jacopo Di Clemente**[4], **Frank E Fish**[5], **Ari S Friedlaender**[6], **William T Gough**[1], **Shirel R Kahane-Rapport**[1], **Cláudia Oliveira**[7], **Susan E Parks**[8], **Gwenith S Penry**[9], **Malene Simon**[10], **Alison K Stimpert**[11], **David N Wiley**[12], **KC Bierlich**[13], **Peter T Madsen**[14,15], **Jeremy A Goldbogen**[1]

[1]Hopkins Marine Station of Stanford University, Pacific Grove, United States; [2]Saint Louis University, St Louis, United States; [3]Cascadia Research Collective, Olympia, United States; [4]Accademia del Leviatano, Rome, Italy; [5]West Chester University, West Chester, United States; [6]Institute of Marine Sciences, University of California, Santa Cruz, United States; [7]Okeanos R&D Centre and the Institute of Marine Research, University of the Azores, Horta, Portugal; [8]Department of Biology, Syracuse University, Syracuse, United States; [9]Institute for Coastal and Marine Research, Nelson Mandela University, Port Elizabeth, South Africa; [10]Department of Birds and Mammals, Greenland Institute of Natural Resources, Nuuk, Greenland; [11]Moss Landing Marine Laboratories, San Jose State University, San Jose, United States; [12]Stellwagen Bank National Marine Sanctuary, Scituate, United States; [13]Duke University Marine Laboratory, Piver's Island, United States; [14]Aarhus Institute for Advanced Studies, Aarhus University, Aarhus, Denmark; [15]Zoophysiology, Department of Bioscience, Aarhus University, Aarhus, Denmark

**\*For correspondence:**
psegre@stanford.edu

**Competing interests:** The authors declare that no competing interests exist.

**Abstract** The considerable power needed for large whales to leap out of the water may represent the single most expensive burst maneuver found in nature. However, the mechanics and energetic costs associated with the breaching behaviors of large whales remain poorly understood. In this study we deployed whale-borne tags to measure the kinematics of breaching to test the hypothesis that these spectacular aerial displays are metabolically expensive. We found that breaching whales use variable underwater trajectories, and that high-emergence breaches are faster and require more energy than predatory lunges. The most expensive breaches approach the upper limits of vertebrate muscle performance, and the energetic cost of breaching is high enough that repeated breaching events may serve as honest signaling of body condition. Furthermore, the confluence of muscle contractile properties, hydrodynamics, and the high speeds required likely impose an upper limit to the body size and effectiveness of breaching whales.

## Introduction

The interface between air and water represents a major barrier for most organisms. The physical characteristics of its supporting medium influence multiple aspects of an animal's physiology, resulting in highly divergent functional adaptations between environments (**Denny, 1993**). Despite the physiological and biomechanical challenges, many taxa take short-term excursions across the air-water interface, yielding a wide variety of benefits that include decreased predation (harbor seals: **da Silva and Terhune, 1988**; flying fish: **Fish, 1990**), thermoregulation (fur seals: **Bartholomew and Wilke, 1956**), parasite removal (sunfish: **Abe and Sekiguchi, 2012**; dolphins: **Weihs et al., 2007**), and increased prey availability (gannets: **Machovsky-Capuska et al., 2012**). In addition, many taxa

exhibit much more brief forays across the fluid interface, exemplified by breaching (from water to air) and plunge-diving (from air to water) in marine vertebrates, both of which are associated with unique mechanical challenges. Whereas plunge-diving animals (e.g. gannets, pelicans) use gravity to accelerate downwards but must contend with the high-speed impacts of entering the more dense water (*Chang et al., 2016*), breaching animals must accelerate upwards against gravity and drag, attaining speeds high enough to exit the water into the much less dense air (*Rohr et al., 2002*).

Breaching, or leaping out of the water, is a well-documented behavior exhibited by many different marine vertebrates, including pelagic rays (*Medeiros et al., 2015*), flying fish (*Fish, 1990*; *Park and Choi, 2010*), squid (*O'Dor et al., 2013*) sharks [*Brunnschweiler et al., 2005*; *Johnston et al., 2018*; *Martin et al., 2005*; *Semmens et al., 2019*], and cetaceans (*Fish et al., 2006*; *Waters and Whitehead, 1990*; *Whitehead, 1985a*; *Whitehead, 1985b*). When coupled with high-speed horizontal travel and streamlined re-entry, low-angle breaching can be further classified as porpoising (*Weihs, 2002*), a behavior that is frequently observed in dolphins and pinnipeds. For small cetaceans traveling at high speeds, porpoising may decrease the cost of locomotion compared to submerged swimming (*Au et al., 1988*; *Weihs, 2002*). In contrast, large whales are rarely if ever observed porpoising, which may suggest that swimmers of this size either face high energetic costs or gain little hydrodynamic benefit from this behavior. Yet, low and high-angle breaching is commonly performed by many species of large whales (summarized in *Whitehead, 1985b*; *Würsig and Whitehead, 2009*). The reasons why large whales breach remain unclear, with possible, non-exclusive explanations ranging from ectoparasite removal (as seen in dolphins, *Fish et al., 2006*) to play (juvenile whales breach frequently, *Würsig et al., 1989*). Another, commonly held explanation is that in large whales, aerial displays are a form of social communication (*Kavanagh et al., 2017*; *Waters and Whitehead, 1990*; *Whitehead, 1985a*), since species with complex social structures breach frequently (e.g., humpback, right, and gray whales), aerial behaviors in humpback whales increase when groups of whales merge or split (*Whitehead, 1985b*), and breaching increases in noisy conditions (*Dunlop et al., 2010*; *Whitehead, 1985a*). In this capacity, breaching may also function as a form of honest signaling to mates and competitors, particularly if the energetic cost of breaching is high.

Large whales generally breach by emerging from the water at a near vertical angle before crashing back down to the surface (*Figure 1*; *Whitehead, 1985b*). However, there is significant variability in breaching behaviors, including different levels of emergence, different exit angles relative to the

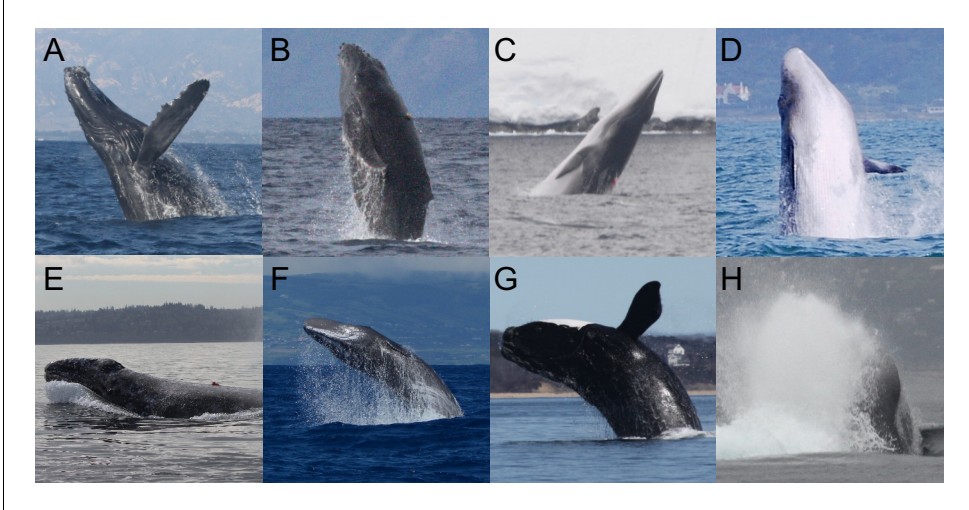

**Figure 1.** Breaching whales. (**A**) A tagged humpback whale (NMFS permit #16111). (**B**) A tagged humpback calf (NMFS permit #14682). (**C**) A tagged minke whale (NMFS permit #14809). (**D**) An untagged Bryde's whale breaching (credit K. Underhill, Simon's Town Boat Company). (**E**) A tagged gray whale falling back into the water (NMFS permit #16111). (**F**) An untagged sperm whale (permit #49/2010/DRA). (**G**) A tagged right whale (MMPA permit #775–1875). (**H**) An untagged blue whale partially emerging from the water while participating in a 'racing behavior' (NMFS permit #16111).

water, and different amounts of long axis-rotation. Most of what we know about breaching comes from above-water performance and observations (*Waters and Whitehead, 1990*; *Whitehead, 1985a*; *Whitehead, 1985b*). Using Lang's (1966) model of dolphin jumping, *Whitehead (1985a)* estimated the speeds of humpback whales immediately prior to a breach as a function of percent emergence from the water and animal length. For angles of emergence greater than 30 degrees, minimum speeds to produce 40% emergence were 1.8 m/s for a 6 m long calf and 2.5 m/s for a 12 m long adult. By relating speed and percent emergence relationships to photographs of breaches, Whitehead calculated a distribution of minimum velocities that preceded the breach, ranging from approximately 1 m/s to 8 m/s. Yet, little else is known about the underwater trajectories used for breaching, how underwater breaching performance compares within and across species, and what the energetic costs of breaching are.

Not all species of large whales breach regularly, and the reasons for this remain unclear. Humpback whales, which can attain body masses greater than 45,000 kg (*Lockyer, 1976*), are frequently observed breaching. The largest species of whales rarely breach: blue whales and sei whales almost never breach (*Whitehead, 1985b*), while fin whales breach rarely and frequent breaching may be confined to specific populations (*Marini et al., 1996*). Likewise, large male sperm whales breach very infrequently while the much smaller females are known to regularly breach (*Waters and Whitehead, 1990*). In concert, these observations suggest that body size may limit breaching performance. One possibility is that the considerable expenditure needed for the largest of whales to accelerate out of their medium may represent too high an energetic cost. Whitehead roughly estimated that during a breach, average sized humpback whales (*Whitehead, 1985a*) and female sperm whales (*Waters and Whitehead, 1990*) expend 1% of their minimum daily basal metabolic requirements. However, little is known about the scaling of breaching energetics and if the cost of breaching increases with size. Alternatively, but not exclusively, body size may impose physical limitations on the swimming capabilities of the largest whales that do not allow them to attain the accelerations or speeds required to breach. Due to the different scaling trajectories of the propulsive surface areas (that generate lift and thrust) and body mass (that resists acceleration), increased body size should decrease accelerative performance (*Webb and De Buffrénil, 1990*).

In this study we used whale-borne tags equipped with inertial sensors to quantify the kinematics of breaching and address the following questions: (1) What are the underwater trajectories and fluking patterns that different species of large whales use to perform breaches? (2) What are the energetic costs of breaching, and how do they scale with body size? And (3) Do energetic or physical constraints impose fundamental limits on the breaching behaviors of large whales? At the upper extremes of body size, the energetic cost of breaching may be prohibitively high. Alternatively, the physical limitations of muscle contractile properties and hydrodynamics may make breaching physically impossible for the largest of whales.

## Results

### Kinematics of breaching

We recorded a total of 187 breaches (*Figure 1*, *Table 1*) from 28 individual humpback whales (n = 152), two minke whales (n = 22), one Bryde's whale (n = 2), one gray whale (n = 1), three sperm whales (n = 6), and two right whales (n = 4). 125 of the breaches were classified as 'full breaches', where > 40% of the whale emerged from the water (*Whitehead, 1985b*); 52 of the breaches were classified as 'partial breaches' (<40% emergence); and 10 were undetermined. The majority of breaches in our dataset were recorded from 28 humpback whales (152 breaches), including three juveniles which were the most prolific breaching whales in our study (106 breaches). For one of the juvenile whales, the shortest time between consecutive breaches was 6.5 s. Humpback whale breaches were highly variable (*Figure 2*, *Figure 3*), with the start of the upward acceleration ranging in depth from 4 m to 52 m and using a variety of trajectories (*Table 2*). Humpback whale breaches featured a wide range of exit speeds (1.1–8.9 m/s), exiting pitch angles (14°−82°), exiting roll angles (2°−178° left or right), and emergence percentages (105 full, 39 partial breaches). At the beginning of the ascent, the flippers are extended to an elevated and protracted position for steering and stability (*Segre et al., 2019*).

**Table 1.** Performance and kinematics of breaching whales.

Mean ± standard deviation are presented along with maximum and minimum values, shown in parentheses. It was not always possible to measure all of the metrics for each breach. Velocity for the gray whale and Bryde's whale breaches were measured using the accelerometer vibrations, while all other velocities were measured using the orientation corrected depth rate.

| | Humpback whale | Humpback juvenile | Minke whale | Bryde's whale | Gray whale | Sperm whale | Right whale |
|---|---|---|---|---|---|---|---|
| # individuals | 25 | 3 | 2 | 1 | 1 | 3 | 2 |
| # events (full, partial breaches) | 46 (39, 6) | 106 (66, 33) | 22 (11, 10) | 2 (2, 0) | 1 (1, 0) | 6 (5, 0) | 4 (1, 3) |
| depth (m) | 24 ± 12 (4, 52) | 9 ± 8 (2, 54) | 7 ± 5 (2, 21) | 12 ± 1 (12, 13) | 5 | 20 ± 6 (12, 29) | 21 ± 11 (10, 31) |
| duration (s) | 7.9 ± 2.3 (4.4, 13.7) | 5.2 ± 2.4 (1.9, 17.6) | 7.5 ± 3.8 (2.9, 18.2) | 5.3 ± 2.2 (3.8, 6.9) | 7.9 | 7.3 ± 1.8 (5.0, 10.2) | 8.8 ± 2.2 (6.9, 11.5) |
| # strokes | 4.1 ± 1.5 (1.7, 6.7) | 2.8 ± 1.6 (1.1, 10.7) | 3.8 ± 1.8 (1.7, 7.5) | - | 2.8 | 3.8 ± 1.2 (2.1, 5.6) | 3.6 ± 1.7 (2.0, 5.4) |
| stroke frequency (Hz) | 0.4 ± 0.1 (0.2, 0.7) | 0.5 ± 0.2 (0.2, 1.1) | 0.5 ± 0.1 (0.3, 0.7) | - | 0.3 | 0.5 ± 0.1 (0.3, 0.6) | 0.4 ± 0.1 (0.3, 0.4) |
| exit speed (m/s) | 6.1 ± 1.8 (2.6, 8.9) | 3.6 ± 1.4 (1.1, 7.6) | 2.7 ± 0.6 (1.6, 3.4) | 5.3 ± 0.6 (4.8, 5.7) | 3.7 | 5.4 ± 1.1 (4.2, 6.5) | 3.0 ± 0.8 (2.2, 3.8) |
| exit pitch (°) | 56 ± 13 (14, 80) | 52 ± 13 (19, 82) | 52 ± 10 (26, 66) | 42 ± 25 (24, 59) | 23 | 49 ± 18 (20, 70) | 49 ± 14 (36, 68) |
| exit roll (°) | 119 ± 57 (4, 178) | 84 ± 58 (2, 179) | 132 ± 39 (37, 177) | 83 ± 116 (1, 165) | 4 | 88 ± 37 (39, 140) | 80 ± 67 (2, 163) |
| emergence (%) | 63 ± 19 (26, 100) | 55 ± 23 (20, 120) | 39 ± 9 (20, 53) | 68 ± 24 (51, 85) | 58 | 65 ± 13 (49, 82) | 33 ± 9 (24, 46) |

Although there is notable flipper movement during the course of the breach it is not clear whether this represents propulsive flapping (*Segre et al., 2017*) or is a stabilizing reaction to the fluke strokes. Breaches can be further characterized by how the whale exits the water, right-side up or upside-down. The videos show that if the whale emerges from the water right-side up, it may arch its back to attain a more vertical position than the shallow exit angle may imply. The videos further suggest that there are two ways that the whale can emerge from the water in an upside-down orientation: (1) the whale does a 'backflip' by increasing its pitch angle past the vertical, or (2) the whale performs a long-axis roll prior to exiting the water. These two maneuvers are not mutually exclusive and can be used together. We did not directly measure rolling velocity, due to the limitations of an accelerometer-based orientation framework. However, the on-board videos suggest that, when employed, rolling can be initiated at different times. With shallow trajectories, the roll is often initiated immediately before the whale breaks the surface of the water: the extended flippers rotate contra-laterally and the whale spins about its long axis. With deeper trajectories, the roll can be initiated much earlier. In both cases, the angular momentum continues the roll after the whale breaks the surface of the water (*Fish et al., 2006*).

We recorded 22 breaches from two minke whales, all of which had shallow U-shaped and shallow V-shaped trajectories (depth 2–21 m, average 7 m, *Table 2*). For the majority of the U-shaped breaches the whales moved at high speeds just below the surface with a last minute upward pitching maneuver, followed by a roll, to take them out of the water (*Figure 4A* shows a deeper version of this maneuver). During these maneuvers, the maximum speed occurs while the whale is moving horizontally and the whale slows once the upward pitching begins. Exit velocities were relatively low (1.6–3.4 m/s) and so emergence percentage was also low (11 full, 10 partial breaches). In 19 of the breaches the whale emerged from the water upside down (roll >90° to either side) and from the videos this seems to come from a combination of backflips (pitching past vertical) and rolls.

The two recorded breaches from Bryde's whales came from a single individual. Both breaches featured high emergence levels and distinctive V-shaped trajectories with the whale starting at the surface and quickly diving to 12 m before pitching upwards and initiating the ascent (*Figure 4B*). The high velocities (4.8 m/s and 5.7 m/s) started before the previous surfacing and were maintained throughout the descent and ascent. One breach had a very low exit angle (24°), while the other had

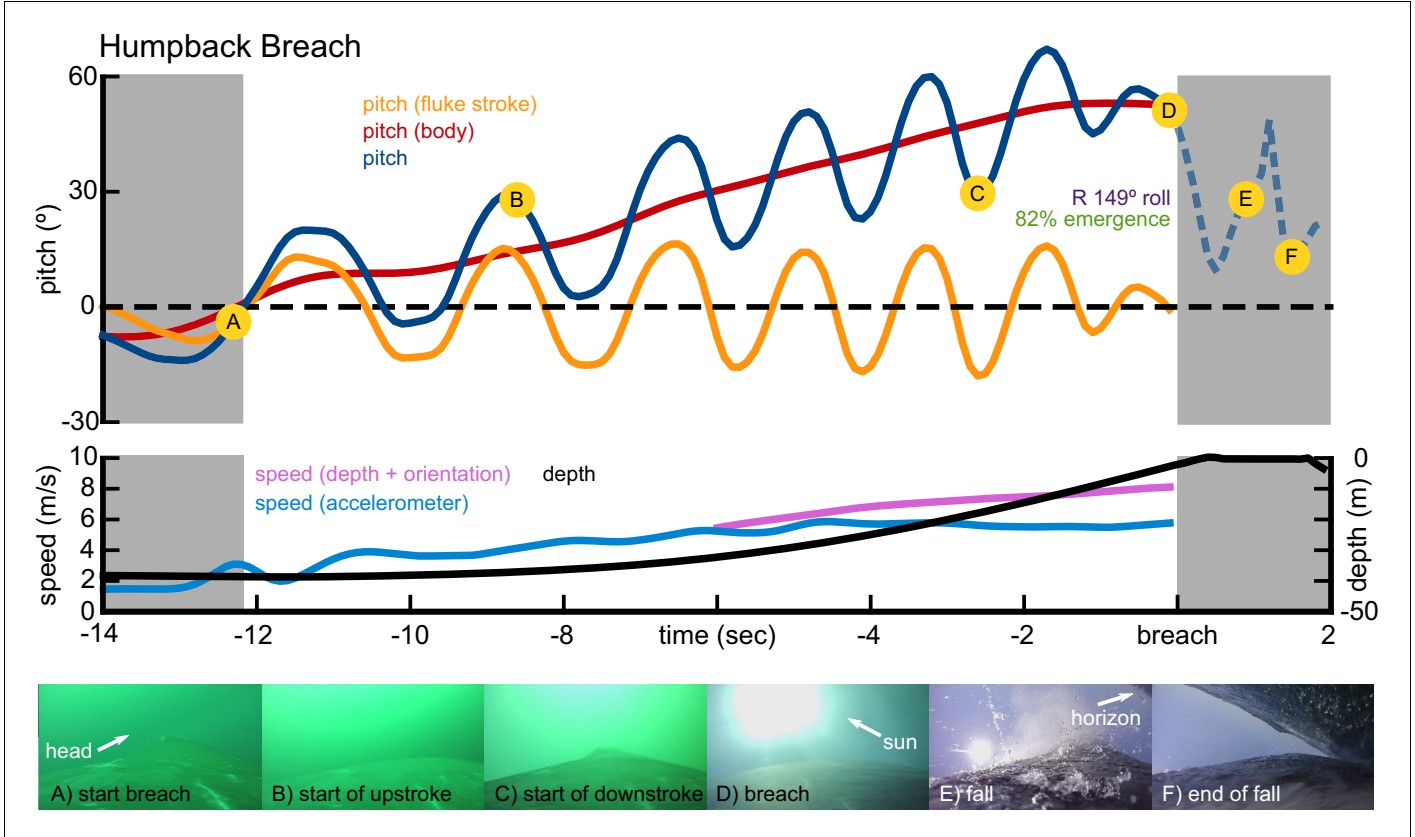

**Figure 2.** Representative breaching kinematics of a humpback whale. Three metrics of pitch are shown: the pitch changes of the body (red), pitch oscillations due to the fluke stroke (orange), and the sum of the two (blue). Two measurements of speed are shown: speed calculated from orientation corrected depth rate (purple), and speed calculated from the accelerometer vibrations (blue). Depth is also shown (black). Images from the onboard camera are shown at specific landmarks during the breach. The video of this breach is included in the supplementary materials (*Video 1*).

a relatively high exit angle (59°), a steeper ascent rate, and the whale emerged upside-down (177° roll), probably having done an underwater backflip during the ascent. These breaches occurred at dusk and were not captured using the on-board cameras so we do not know if the whale rolled while exiting the water.

In the single gray whale breach that we recorded, the whale dove to 5 m, swam horizontally at a high speed, and performed a quick upward pitch to emerge from the water at 3.7 m/s (U-shape, *Figure 4C*). The exit angle was low (23°) but in the video the whale distinctly arched its back as it emerged (a full breach) with an upright roll angle (4°). This breach was likely a response to tagging, since it occurred immediately after the deployment.

We recorded six breaches from three female or juvenile sperm whales. Five breaches had a V-shaped trajectory with the whale descending between 12 m and 29 m before turning upwards and beginning the rapid ascent (*Figure 4D*). One breach had a J-shaped trajectory with the whale ascending slowly without fluking before clearly beginning its rapid acceleration. The maximum recorded velocity was 6.5 m/s but all the breaches were fast (avg 5.4 m/s). Five of the breaches were full breaches (one was indeterminate) with variable exit pitch angles (20° - 70°) and roll angles (39° - 140°). We do not know if the whales performed rolls or backflips when they emerged upside-down.

Finally, we recorded four breaches from two right whales. All of the breaches had relatively slow exit velocities (maximum 3.8 m/s) with low levels of emergence (one full; three partial breaches). The two V-shaped dives had slower exit speeds (2.2 m/s and 2.6 m/s), shallower depth (13 m and 10 m), and were both partial breaches. The two I-shaped dives (*Figure 4E*) began with the whale holding station at ~30 m before beginning a rapid, direct upward acceleration. These breaches featured higher exit velocities (3.6 m/s, 3.8 m/s), higher levels of emergence (one full; one partial), and one of

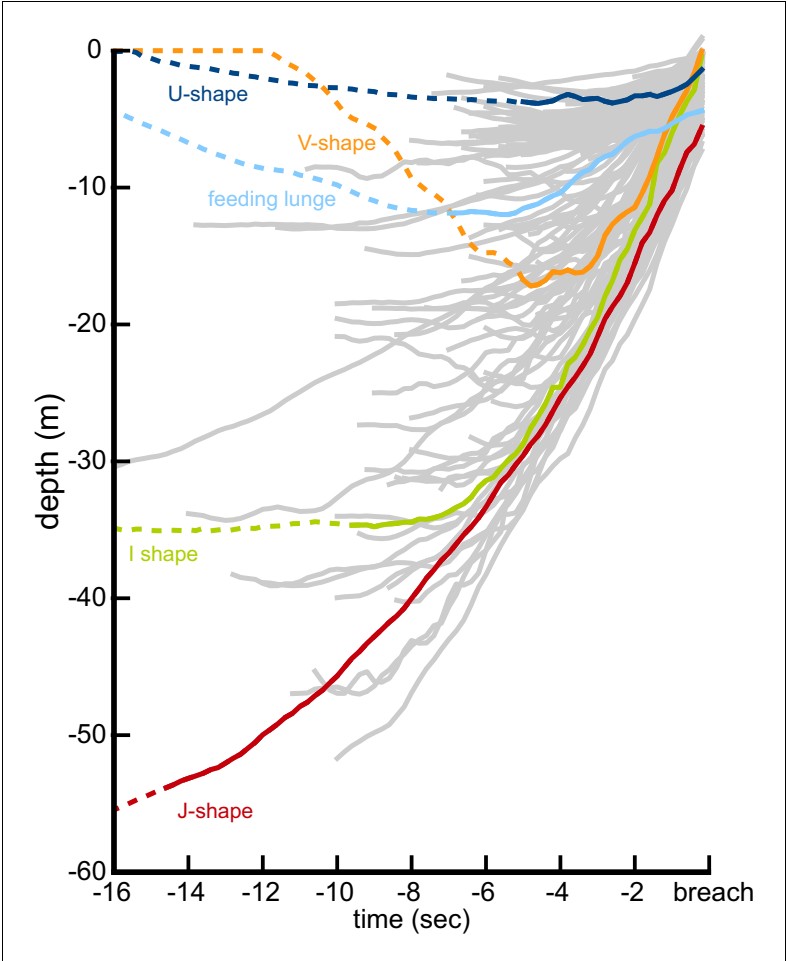

**Figure 3.** The diversity of underwater breaching behavior is illustrated by the depth profiles of 152 breaching accelerations performed by 37 humpback whales. Four representative trajectories illustrating U, V, I, and J-shaped breaching profiles are highlighted, showing both the beginning of the upwards acceleration (solid line) and the 16 s prior to the breach, provided for context (dotted line).

the whales emerged upside-down (163°). Both whales were tagged in relatively shallow water (~30 m) which may have constrained their breaching performance.

Across all breaches there was a strong positive correlation between exit speed and starting depth ($R^2 = 0.67$), with an extra 1 m/s gained for every four additional meters of depth (*Figure 5A*). There was also a strong correlation between exit speed and average stroke frequency ($R^2 = 0.72$) and there are clear differences between the smaller animals (minke whales, juvenile humpback whale) and the larger animals (*Figure 5B*). There was no correlation between exit speed and exit pitch angle (*Figure 5C*). There were few clear relationships between exit speed and exit roll except that adult humpback whales and minke whales often emerged from the water upside-down (>90° roll, *Figure 5D*).

## Energetics of breaching

The energetic costs of breaching were calculated for five humpback whales with known body dimensions and high-emergence breaches. Both the total energy expended and maximum mechanical power required to breach increased with body mass (*Equations 25-27*; *Table 3*). The mass-specific energetic cost of breaching also increased with body mass (range: 7000 kg, 130 kJ/kg to 46000 kg, 220 kJ/kg; *Table 3*; *Figure 6A*). This increase in energetic expenditure was driven by the increase in breaching speed with mass (range: 6.2 m/s to 8.2 m/s; *Table 3*; *Figure 6B*), and the mass-specific power output required to attain these higher speeds also increased with body mass (range: 7 W/kg

**Table 2.** Breaching trajectories were broadly categorized based on their shape.

| Trajectory | Starting location | Characteristics | Species | # events |
|---|---|---|---|---|
| U-shape | surface | horizontal acceleration slightly below the surface; rapid upward pitch change to emerge from water (*Whitehead, 1985a*) | humpback | 1 |
| | | | humpback, juv. | 80 |
| | | | minke | 17 |
| | | | grey | 1 |
| V-shape | surface | powered or unpowered descent; abrupt, upward change of direction to start ascent | humpback | 21 |
| | | | humpback, juv. | 18 |
| | | | minke | 4 |
| | | | Bryde's | 2 |
| | | | sperm | 5 |
| | | | right | 2 |
| J-shape | depth | slow ascent from depth; abrupt rapid acceleration towards surface | humpback | 4 |
| | | | humpback, juv. | 4 |
| | | | sperm | 1 |
| I-shape | depth | holding station at depth; abrupt, rapid acceleration towards surface | humpback | 20 |
| | | | humpback, juv. | 4 |
| | | | minke | 1 |
| | | | right | 2 |

to 11 W/kg; *Figure 6C*). Rorqual whales feed by rapidly accelerating, opening their mouths, and engulfing large volumes of prey-laden water. Although the trajectories used for feeding lunges are highly variable (*Cade et al., 2016*; *Simon et al., 2012*), lunges are common behaviors that require a rapid acceleration similar to that used for breaching. For each of the five humpback whales, the cost of breaching was higher than the cost of accelerating to perform their highest-speed lunge.

Relative to daily Field Metabolic Rate ($FMR_{daily}$), the cost of breaching increased with increasing mass and was always higher than the cost of accelerating for a high-speed feeding lunge (*Supplementary file 1A*). This pattern held regardless of which equation was used for predicting $FMR_{daily}$ of large whales. However, the *Williams and Maresh (2015)* equation for scaling of $FMR_{daily}$ resulted in a higher cost of breaching (*Equation 28*; range: 0.5% to 2.3% of $FMR_{daily}$) than the modified *Nagy (2005)* equation (*Equation 29*; 0.08% to 0.20% of $FMR_{daily}$).

## Discussion

The considerable power needed for large whales to leap out of the water may represent the single most expensive burst maneuver found in nature. However, the mechanics and energetic costs associated with the breaching behaviors of large whales remain poorly understood. In this study we first examined the underwater trajectories that large cetaceans use for breaching to determine if historical hypotheses about underwater movement were correct. Next, we used a hydrodynamic model to estimate the energetic costs of breaching and how it scales with body size. It has been hypothesized that extended breaching sequences can serve as an honest signal of fitness (*Whitehead, 1985b*); however, this depends on whether breaching is an energetically expensive behavior. Finally, we test the hypothesis that energetic or physical constraints

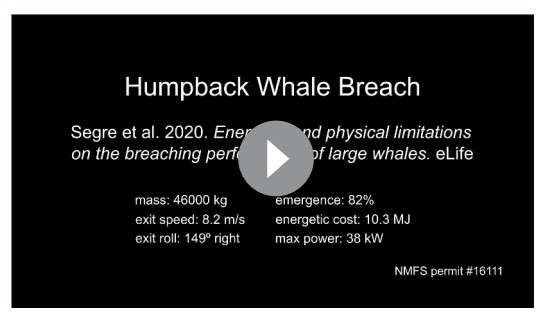

**Video 1.** Camera-tag video of a humpback whale performing a breach. The trajectory of this breach is shown in *Figure 2*.
https://elifesciences.org/articles/51760#video1

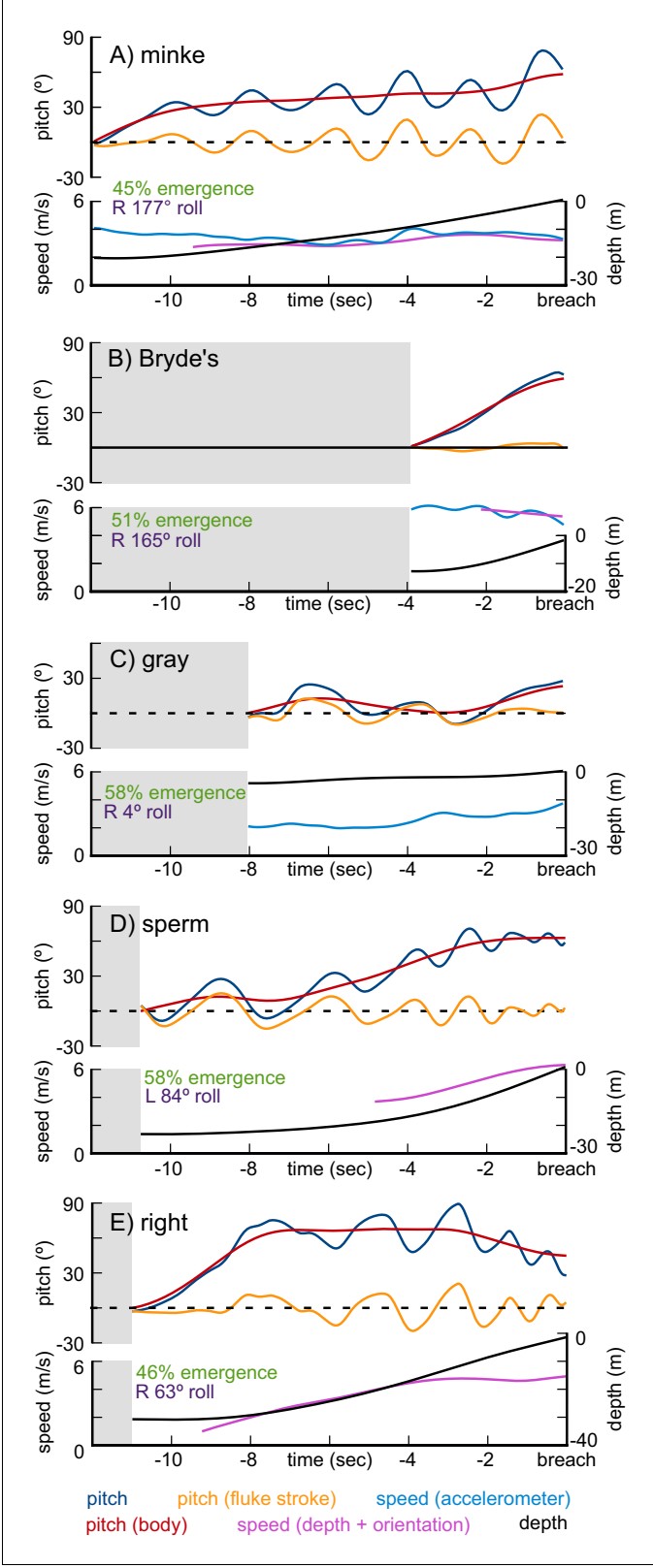

**Figure 4.** Representative breaching kinematics of a minke whale (**A**), a Bryde's whale (**B**), a gray whale (**C**), a sperm whale (**D**), and a right whale (**E**). Three metrics of pitch are shown: the pitch changes of the body (red), pitch oscillations due to the fluke stroke (orange), and the sum of the two (blue). Two measurements of speed are shown: speed calculated from orientation corrected depth rate (purple), and speed calculated from the

*Figure 4 continued on next page*

*Figure 4 continued*
accelerometer vibrations (blue). Depth is also shown (black). The graphs show the 12 s before the whale emerges from the water, with gray shaded areas representing time before the breaching maneuver begins.

impose fundamental limits on the breaching behaviors of the largest whales. It is possible that for large whales the energetic cost of breaching is prohibitively high. Alternatively, it may be hypothesized that physical limitations of muscle contractile properties and hydrodynamics constrain the effectiveness of breaching in the largest of animals.

## How do large whales breach?

The underwater trajectories that allow whales to leap out of the water have been the subject of much speculation, largely because the bio-logging equipment that makes the quantitative study of underwater locomotor performance possible has only recently been developed and widely adopted (*Goldbogen et al., 2017*; *Johnson and Tyack, 2003*). Our data show that the underwater breaching trajectories are variable, even within species. Whitehead (*Whitehead, 1985b*) described humpback whale breaching trajectories as having a shallow horizontal approach before pitching-up and leaving the water, and Payne described similar trajectories for right whales (see *Waters and Whitehead, 1990*). We did find many examples of this trajectory in humpback and right whales, and we also found this trajectory used by minke and gray whales. In addition, it has been suggested that sperm whales require long ascents to breach (70–110 m; *Whitehead, 2003* p. 176), but we demonstrate that they can breach even from relatively shallow depths (12–29 m) using only a few fluke strokes (2–6 strokes). We also found that humpbacks, minkes, sperm, and right whales used other types of trajectories while breaching: starting at the surface and diving, holding station, and ascending to the starting depth before beginning the breaching ascent. We had too few breaches from Bryde's whales and gray whales to uncover any diversity in the trajectories. We did find support for Whitehead's observations that adult humpback whales generally emerge right-side up or upside-down (*Figure 5D*), although we found some adults that emerged on their sides. Our video data suggest a mechanism for this pattern: adult humpback whales appear to incorporate less long-axis angular velocity into their breaching trajectories. Instead, they often emerge right-side up or pitch upwards, past vertical and emerge upside-down. In contrast, juvenile humpback whales often leave the water with a distinct rolling velocity, which results in a more unpredictable roll angle as they emerge (*Figure 5D*). Since both adults and juveniles often rotate their flippers contra-laterally before emerging, it is not clear whether the difference is behavioral or the result of the larger adults having to overcome their higher rotational inertia.

## Maximal swimming performance during breaching events

Breaching events can uniquely shed light on maximal locomotor performance of large animals, at the extremes of body size, which is a topic that has remained elusive (*Gough et al., 2019*). For most of the species examined in this study, our ability to discuss maximal performance is influenced by low sample sizes. However, for humpback whales we measured large numbers of breaches (152) from many individuals (28), and data from our fastest breaches match well with previous observations and theoretical predictions. Most data on the maximal swimming speeds of rorquals have been anecdotal (*Hirt et al., 2017*), relying on observations of whales as they swam away from moving boats. *Lockyer (1981)* reported that humpback whales could swim up to 7.5 m/s when alarmed. Using speeds calculated from photographs of humpback whales breaching, *Whitehead (1985a)* reported a top speed of 8.2 m/s, although he suggested that this may have been an overestimate. Both of these estimates were very close to our results: we measured seven breaches from seven individual adult humpback whales which achieved top breaching exit speeds of over 8 m/s, with a maximum of 8.9 m/s. Our examination of humpback whales with known body lengths (and calculated body masses) registered accelerations ranging from 0.5 to 0.75 $m/s^2$, and suggests that top swimming speed increases (*Table 3*; *Figure 6B*) and stroke frequency decreases (*Table 3*) with body size.

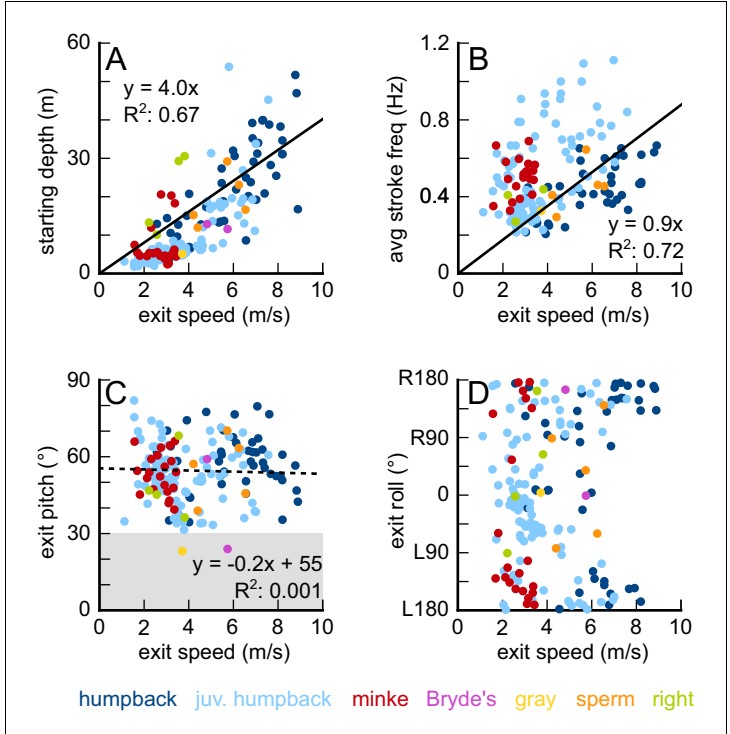

**Figure 5.** Breaching speed is correlated with starting depth (**A**) and average stroke frequency (**B**), but not with breaching pitch (**C**), or breaching roll angle (**D**).

The online version of this article includes the following source data for figure 5:

**Source data 1.** Data from 187 breaches performed by 28 individual humpback whales, two minke whales, one Bryde's whale, one gray whale, three sperm whales, and two right whales.

## Is breaching energetically expensive?

In absolute terms, the amount of energy required for a large whale to leap out of the water is extraordinary. For a 7.8 m humpback whale, the cost of performing a single full breach is 0.9 MJ but for a 14.8 m whale the cost increases to 10.3 MJ (*Table 3*), which is equivalent to the energy required for a 60 kg runner to complete a marathon (*Margaria et al., 1963*). Furthermore, because breaches happen so quickly, the mechanical power required to breach is also extremely high. The second largest humpback whale in this study (14.7 m, 46,000 kg) produced an average mechanical power output of 300 kW over the course of its 8.5 s breach, or approximately the maximum pulling power of 25 draft horses (*Collins and Caine, 1926*). The energetic expenditure of this breach was also roughly equivalent to the energetic cost of the largest blue whale in our database performing

**Table 3.** Kinematic and energetic parameters for five breaches and five high performance lunges performed by five humpback whales spanning a range of sizes.

| Length (m) | Mass (kg) | Emergence (%) | Duration (secs) | | Final velocity (m/s) | | Stroke freq (Hz) | | Energy (MJ) | | Max power (kW) | |
|---|---|---|---|---|---|---|---|---|---|---|---|---|
| | | Breach | Breach | Lunge | Breach | Lunge | Breach | Lunge | Breach | Lunge | Breach | Lunge |
| 7.8 | 7000 | 86 | 8.0 | 6.8 | 6.2 | 5.3 | 0.7 | 0.5 | 0.9 | 0.7 | 5 | 5 |
| 10.5 | 17000 | 79 | 8.1 | 4.7 | 7.1 | 5.0 | 0.6 | 0.4 | 2.8 | 1.2 | 15 | 10 |
| 12.7 | 30000 | 61 | 9.1 | 2.9 | 6.0 | 5.0 | 0.4 | 0.3 | 3.7 | 1.6 | 23 | 18 |
| 14.7 | 46000 | 84 | 8.5 | 3.3 | 8.2 | 4.8 | 0.5 | 0.3 | 9.8 | 2.6 | 50 | 25 |
| 14.8 | 46000 | 82 | 12.7 | 6.1 | 8.1 | 5.4 | 0.5 | 0.2 | 10.3 | 3.6 | 38 | 23 |

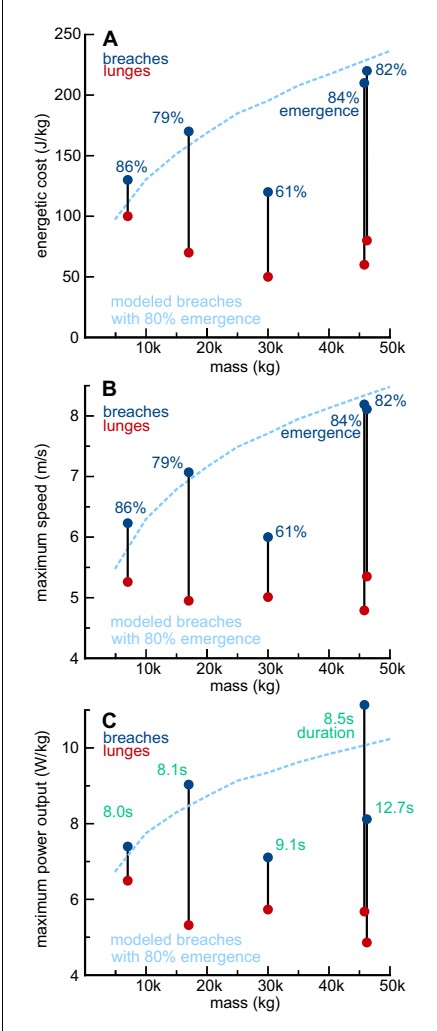

**Figure 6.** The cost of breaching increases with body size, in humpback whales. (**A**) The mass-specific energy expenditure required to perform high-emergence breaches (blue) and high-performance lunges (red) is shown for five humpback whales of different sizes. Because the whales breached with different percentages of their bodies emerging from the water (dark blue numbers), the expected relationship between mass and the energetic cost of breaching with 80% body emergence, is shown for comparison (light blue line). The modeled breaches were calculated using average parameters from the trajectories of the five individuals shown (65° pitch; body width = 18% of length; 1.75 m/s starting velocity; 0.65 m/s² acceleration; no plateau phase). Both the model and the data show that the mass-specific cost of breaching increases with body size. (**B**) This pattern is largely driven by the higher speeds that larger whales need to emerge from the water. (**C**) To attain the higher speeds required to emerge from the water, larger whales need to generate higher mass-specific mechanical power outputs or extend the duration of their trajectories (green numbers).

its fastest lunge (25.2 m, 5.7 m/s, 9.5 MJ, unpublished data) in spite of the humpback whale having half the mass of the blue whale. Thus a breach is much more energetically expensive than a high-speed predatory lunge.

In relative terms, the cost of breaching is less clear. If the relationship between body mass and field metabolic rate proposed by *Williams and Maresh (2015)* holds for larger cetaceans, then increased size comes with high metabolic efficiency and the daily field metabolic rate is low. This, in turn, makes breaching relatively expensive: humpback whales may spend between 0.5% and 2.3% of their daily energy budget performing a single full breach. For juvenile humpback whales the cost of performing a single breach represents a smaller percentage of their average daily energy budget (0.5%), a number which is slightly more expensive than a high-performance feeding lunge (0.4%). However, for large adults (46,000 kg) the cost of a single breach increases to 2.3% of $FMR_{daily}$ and is substantially higher than the cost of a single high-performance feeding lunge (0.8%). On the other hand, if the scaling relationship between body mass and $FMR_{daily}$ is closer to that of terrestrial animals (e.g., *Nagy, 2005* elevated by 50%, *Rojano-Doñate et al., 2018*), then the daily field metabolic rate is high and humpback whales would spend a substantially lower percentage of their daily energy budget while performing a breach (0.08% to 0.20%). In both scaling scenarios the cost of breaching increases with body mass and this relationship is mostly driven by the increased metabolic efficiency that comes with larger size (*Nagy, 2005*; *Williams and Maresh, 2015*), but it is also partially a result of the increased speed and momentum required for larger animals to emerge from the water (*Figure 6B*). Notably, the relative cost of performing a breach is much lower for humpback whales (0.08–0.5% $FMR_{daily}$ for the small 7000 kg whale) than for basking sharks (5–6% $FMR_{daily}$ for a 2700 kg shark, *Johnston et al., 2018*). Because the physics of breaching remains similar across similarly sized organisms, the low $FMR_{daily}$ that comes with being ectothermic makes breaching relatively much more expensive for sharks. Thus, for basking sharks or white sharks (which use fast, vertical ascents to target prey near the surface; *Semmens et al., 2019*), a single breach may represent a very expensive event, while humpback whales can perform multiple breaches before the costs begin to accumulate.

Many of the individual whales we tracked performed multiple, sequential breaches. One

juvenile humpback performed at least 69 breaches and a series of other aerial behaviors over the course of two days (17 during a 6.75 hr deployment on the first day; 52 during a 4.5 hr deployment on the second day). In many animals, the energetic cost of performing even trivial, but frequently repeated behaviors can be substantial (*Dudley and Milton, 1990*). Regardless of which scaling regime is used to calculate metabolic rates, the cost of repeated breaching represents a significant energetic expenditure for whales. While at their calving grounds, capital breeding females in a fasting state maintain low metabolic rates in order to devote most of their energy to nursing their calves (*Bejder et al., 2019*). In spite of this, repeated breaching is commonly observed, often with the mothers and calves breaching side-by-side. Thus, the energy expended breaching cannot be put towards lactation (for mothers) or storing blubber (for the calves). Unlike feeding lunges, which are relatively less expensive but are also used to acquire energy, the cost of breaching on the breeding grounds will not be recouped until the whales return to their feeding grounds, several months later (*Christiansen et al., 2016*). This suggests that repeated breaching has a social purpose important enough to warrant the high energetic expense, perhaps serving a developmental function for juveniles or an honest signal of fitness for adults.

## Does body size limit breaching performance?

On a mass-specific basis, the cost of breaching also increases with body size (*Figure 6A*) and this increase is largely driven by the higher speeds required to emerge from the water (*Figure 6B*). In turn, the locomotor muscles must generate higher power outputs to accelerate to these higher speeds (*Figure 6C*), even though maximum mass-specific force production decreases with body size (*Arthur et al., 2015*). This suggests that there may be an upper size limit to breaching ability based on the limitations of muscle power-generating capabilities. The mass-specific power outputs that we measured during the last second of each breaching acceleration are all slightly lower than previously reported values for smaller cetaceans swimming at high speeds (22 W/kg - 31 W/kg, *Fish, 1998*). Additionally, the second largest humpback whale of our study generated approximately 85 W/kg of locomotor muscle mass (~13.2% of body mass, *Arthur et al., 2015*) during the last second of its acceleration. Although little is known about power-generating capabilities of cetacean muscles, this value is near the limits muscle performance in other vertebrate taxa (*Jackson and Dial, 2011*; *Marden, 1994*). Since power is time dependent, a large whale could decrease its power requirements by extending the length of its breaching trajectory, which explains some of the variation in *Figure 6C*. The largest whale in this analysis took a long time (12.7 secs vs 8.5 secs for the second largest whale) to accelerate slowly (Table S1B) to its exit speed, expending more energy but decreasing its maximum power output (*Table 3*). However, this strategy likely has its limits, since the duration of a trajectory may be constrained by the onset of muscle fatigue. Our model (blue line, *Figure 6A–C*) suggests that the largest of whales would require even higher speeds to emerge from the water, but that their muscles may not be able to generate enough power or sustain a swimming trajectory long enough to attain these speeds.

Why do larger whales require higher speeds to breach? Whitehead's model (1985a) for calculating the emergence percentage for a given breaching speed and exit angle suggests that length is a more of a hindrance to breaching than mass. This is similar to how a projectile thrown upwards reaches its maximum height based solely on its initial velocity, regardless of its weight. Therefore, if our large blue whale (25.2 m) breached using a similar trajectory to our largest measured humpback whale (12.7 s duration; *Table 3*), it would have to swim at 10.9 m/s to emerge with the same percentage, expending approximately four times the energy (40.3 MJ, 0.4–6.3% $FMR_{daily}$) and requiring a higher mass-specific power output (14 W/kg). It is not clear whether blue whales can even reach this speed (*Gough et al., 2019*), which may be limited by both muscular power output and the hydrodynamic limits of lunate tail propulsion (*Iosilevskii and Weihs, 2008*). 'Racing' blue whales reach speeds of approximately 7.5 m/s with faster bursts, often performing very low emergence breaches in the process (*Torres et al., 2017*); J.C. unpublished data, *Figure 1H*), but this is the best estimate of the maximum swimming speed that blue whales can attain. The relationship between length and emergence may also explain why large, rotund species like right, bowhead, and humpback whales breach more often than large slender species, like fin and blue whales (*Whitehead, 1985b*). Right whales and bowhead whales attain large masses due to their rotund shape but are similar in length to humpback whales. In comparison, the largest fin whales are as heavy as the largest right whales, but are also ~50% longer (*Lockyer, 1976*). Sexually dimorphic

male sperm whales are ~50–100% longer and 3–5 times heavier than their female counterparts and do not breach very often. In 59 tag deployments on fin whales, we recorded one breach (which caused the tag to slip before the whale exited the water), while in 14 tag deployments on male sperm whales and in 156 tag deployments on blue whales we recorded no breaches.

The physical and behavioral limitations on breaching performance are likely more complex and nuanced than the first approximations presented here. On an inter-specific level, variation in the scaling of propulsive surfaces (*Woodward et al., 2006*), muscle mass (*Arthur et al., 2015*), and hydrodynamic variation (*Fish and Rohr, 1999*) probably have a strong influence on the maximal locomotor performance required for breaching. Additionally, differences in body-composition and buoyancy may make it easier for certain species to breach (i.e., positively buoyant right whales; *Nowacek et al., 2001*). Intra-specific factors such as body-condition (*Miller et al., 2004*; *Nowacek et al., 2001*) and individual morphological variation (*Kahane-Rapport and Goldbogen, 2018*) may also play a role in limiting breaching performance. Even on an individual level, the amount of air stored in the lungs and the resulting changes in buoyancy (*Miller et al., 2004*) may influence the forces involved during different breaching events. Meanwhile, the physical ability to breach efficiently combined with a complex social structure and high levels of innate maneuverability may have predisposed certain species, such as humpback whales, to incorporate breaching as a form of communication.

In conclusion, our results suggest an underlying biomechanical explanation for the factors that limit intra-specific and inter-specific breaching ability in large whales. We found that breaching whales use variable underwater trajectories, and that high-emergence breaches feature speeds approaching the upper limits of locomotor performance. The speeds required to substantially emerge from the water result in high energetic costs that increase disproportionately with body size. The cost of performing extended breaching sequences certainly represents a significant energetic expenditure, supporting the hypothesis that breaching serves an important social function for some species. However, the energetic cost of performing a single, isolated breach is likely not sufficient to explain why the largest of whales do not breach. Instead, our analysis suggests that the breaching ability of large whales may be limited by the capacity of their muscles to deliver high bursts of power or sustain high-speed trajectories for extended durations. The confluence of muscle contractile properties, hydrodynamic limitations of lunate tail propulsion, and the higher speeds required for longer whales to emerge from the water likely imposes an upper limit to the body size and effectiveness of breaching whales.

## Materials and methods

Between 2009 and 2018 we deployed suction-cup attached bio-loggers on humpback (*Megaptera novaeangliae*; several locations worldwide), minke (*Balaenoptera bonaerensis*; Antarctica), inshore Bryde's (*Balaenoptera edeni*; Plettenberg Bay, South Africa), gray (*Eschrichtius robustus*; Puget Sound, WA), sperm (*Physeter macrocephalus*; Azores), and right whales (*Eubalaena glacialis*; Cape Cod Bay, MA). We used two types of bio-logging tags (DTAG2: *Johnson and Tyack, 2003*; CATS: [*Cade et al., 2016*; *Goldbogen et al., 2017*]) that differed in specifications, but were equipped with depth and temperature sensors (DTAGS: 50 Hz; CATS: 10 Hz), three-axis accelerometers (DTAG: 50 Hz; CATS: 400 Hz), and three-axis magnetometers (DTAG: 50 Hz; CATS: 50 Hz), all 16 bit. The DTAGs were deployed on sperm, right, and humpback whales. The CATS bio-loggers were also equipped with cameras and were deployed on humpback, minke, Bryde's, and gray whales. Bio-loggers were also deployed on three juvenile humpback whales: CATS tags were deployed on two smaller animals in their feeding grounds, and a DTAG was deployed using a special protocol designed to minimize disturbance, on a calf in the breeding grounds (*Stimpert et al., 2012*). We identified breaches (*Figure 1*) by watching the onboard videos (CATS tags, *Figure 2*, *Video 1*), using surface observation data, or manually examining the data for rapid ascents that were followed by sections where the depth sensors abruptly emerged from the water (0 m depth; *Figure 2*). We only included breaches where the suction-cups did not slip throughout the ascent, and where we could confidently estimate the orientation of the tag on the whale (*Johnson and Tyack, 2003*). Deployments that contained breaches represented a small subset of larger datasets collected for different projects.

Once we identified breaching events, the raw data were downsampled to 5, 10, or 25 Hz depending on the original dataset. We applied a zero-lag Butterworth filter designed to remove sampling error from the accelerometer and magnetometer data (low pass, cutoff frequency: 1 Hz) and calculated the orientation of the whale using the standard pitch, roll, and heading framework (*Johnson and Tyack, 2003*). We then applied another series of zero-lag Butterworth filters to the pitch signal to separate the contribution of the body orientation (low pass, cutoff frequency: 0.2 Hz) from the contribution of the fluke strokes (high pass, cutoff frequency: 0.2 Hz) to the overall pitch (*Martín López et al., 2015*). For each breach we identified the start of the maneuver as the time when the body pitched upwards past horizontal and began the ascent towards the surface. In some cases, when the whale was already ascending from a dive, we defined the start of the breaching ascent by manually finding the time when the fluke strokes began or intensified. The depth sensors clearly showed when the tag exited the water, but often the tag placement was distal enough that by the time the tag broke the surface, the whale was already falling out of the air. Therefore, to accurately measure the underwater trajectories associated with breaching, we estimated the time when whale broke the surface, using the depth sensor and the pitch angle as a guide to ensure the whale had not already started its abrupt downward, aerial trajectory. We estimated speed using two methods. (1) At high pitch angles (>30°) we used the orientation-corrected depth rate (*Miller et al., 2004*). This method is only valid at high pitch angles, and was used to calculate most of the exit velocities reported in *Table 1*. (2) For the CATS tag deployments we calibrated the measurements of the background, high frequency accelerometer vibrations (sampled from the 400 Hz data) with the orientation-corrected depth rate (*Cade et al., 2018*). At high speeds this method may underestimate velocity due to clipping of the accelerometer signal, and therefore we only used it to calculate exit speeds of the gray whale and the Bryde's whale breaches, where exiting pitch angles were low. We used a combination of both methods to calculate the velocity profiles of the humpback breaches and lunges used for the energetic analysis.

## Kinematic analysis

The breaching trajectories were broadly classified by shape (*Table 2*). From the breaching data we calculated a series of performance metrics including the depth at the start of the breach, the duration of the breach, the pitch when the whale exited the water, and the roll when the whale exited the water (if the pitch was <80°, to avoid gimbal lock). The sinusoidal fluke strokes were not always visible in the data, particularly when the tag was placed anteriorly. When possible (167 breaches), we counted the number of fluke strokes (upstroke to upstroke or downstroke to downstroke) per breach, by using the zero-crossings of the high-pass filtered pitch signal. We did not include the last half-stroke as the whale emerged, but we did include the part of the first stroke that occurred as the breach began - expressed as a fraction. We calculated the average stroke frequency over the course of the breach.

We also calculated a rough estimate of the percentage of the whale that emerged from the water, using the simple physics-based model from *Whitehead (1985a)* and *Lang (1966)*. We used exit velocities and pitch angles derived from the sensor data, modeling the whales as cylinders. The body length of the whales were estimated using either photos taken from unoccupied aerial vehicles (seven adult humpbacks; one juvenile humpback; one minke) or species averages (adult humpback = 14 m; juvenile or calf humpback = 7 m; minke = 7.8 m; gray = 12 m; female sperm = 11 m; right = 14 m; Bryde's = 13 m; *Lockyer, 1976*). We classified aerial behaviors as full breaches when > 40% of the whale emerged from the water (*Whitehead, 1985a*). The remaining behaviors were classified as partial breaches. When available, video data confirmed these emergence calculations and classification system. Although coarse, this method provides a useful separation between high-performance and low-performance breaches.

To examine the relationships between kinematic variables associated with breaching we used a linear mixed effects model with nested random effects (individuals nested within species). We calculated a pseudo-$R^2$ designed for use with Bayesian regression models: the variance of the predicted values divided by the variance of predicted values plus the variance of the errors (*Gelman et al., 2019*). Statistics were performed using the Statsmodels package in Python.

## Energetic analysis

We estimated the energetic cost of breaching using breaches from five individual humpback whales of different sizes (7.8 m to 14.8 m, as measured by unmanned aerial photogrammetry; *Table 3*; *Durban et al., 2016*; *Johnston, 2019*). For each individual we selected a high-performance breach (60–90% emergence) with a stereotypical acceleration profile (starting at a low speed and rapidly accelerating to the surface). As a comparison, for each individual we also selected the fastest lunge (individuals had between 12 to 342 lunges) with a stereotypical acceleration profile (also starting at low speed and rapidly accelerating; *Figure 3*). We measured the speed at the start of the maneuver using the accelerometer vibration method, because the pitch was often below the 30° threshold required for calculating orientation-corrected depth rate. We measured the velocity at the end of the maneuver using orientation-corrected depth rate to avoid any accelerometer clipping that may occur during the highest accelerations.

The energetics of breaching and lunging were estimated using a two-step process. First, the mechanical work of the system was calculated by adding the work done against drag to the change in kinetic energy. Second, the metabolic energy spent by the muscles to perform the work was estimated using metabolic efficiency coefficients (*Blake, 1983*; *Fish, 1993*; *Fish, 1998*; *Webb, 1971*; *Webb, 1975*). These calculations represent the cost of accelerating and do not include estimates of basal metabolic rate.

## Parameters from bio-loggers and aerial photography

Using data from the bio-loggers, each breach and lunge was split into two phases: an acceleration phase where the velocity increased from the initial velocity ($U_i$) to the final velocity ($U_f$) over the duration of $T_{acc}$ seconds, and a plateau phase where the velocity stayed constant at $U_f$ for the duration of $T_{plat}$ seconds (*Supplementary file 1* - Table S1B). When there was no plateau phase, $T_{plat}$ was set to zero. We did not include costs incurred after breaking the water (for breaching) or after opening the mouth (for lunge feeding), and so this analysis functionally compares the approach phase of a breach to the approach phase of a high-performance lunge. For simplicity we assumed a neutral buoyancy given that the forces involved differ with species, body condition, and air stored in the lungs (*Miller et al., 2004*; *Nowacek et al., 2001*), and remain poorly understood. Body length ($L_{body}$) and maximum body width ($w_{max}$) were estimated from aerial photographs (*Johnston, 2019*). Body mass ($M_{body}$) was estimated from body length using the equations from *Lockyer (1976)*.

## Mechanical energy required for swimming

A moving whale producing thrust by fluking must perform enough mechanical work to overcome drag. The relationship between work performed by fluking ($W_{Thrust}$), the work that is required to overcome drag ($W_{Drag}$), and the change in kinetic energy of the whale is given by the work-energy theorem:

$$\frac{1}{2}M_{body}\left(U_f^2 - U_i^2\right) = W_{thrust} - W_{drag} \tag{1}$$

Rearranged this becomes:

$$W_{thrust} = \frac{1}{2}M_{body}\left(U_f^2 - U_i^2\right) + W_{drag} \tag{2}$$

This equation can be used to calculate the mechanical work produced during either the acceleration phase or the plateau phase. To calculated the total work produced the two are added together. During the plateau phase velocity is constant ($\Delta U = 0$) and so the kinetic energy is zero, leaving:

$$W_{\text{thrust, total}} = \frac{1}{2}M_{body}\left(U_f^2 - U_i^2\right) + W_{\text{drag, acc}} + W_{\text{drag, plat}} \tag{3}$$

The work required to overcome drag is calculated from the time integral:

$$W_{\text{drag, acc or plat}} = \int_{x_{initial}}^{x_{final}} F_{drag}(t)\,dx = \int_{t_{initial}}^{t_{final}} F_{drag}(t)U(t)\,dt \tag{4}$$

where the differential of distance is substituted with the differential of time through the relationship $U(t)=dx/dt$.

## Drag incurred at constant speed

During the plateau phase, velocity is constant ($U(t)=U_f$). The drag force also remains constant over time, since it depends on velocity (as will be shown below), and therefore *Equation 4* becomes:

$$W_{\mathrm{drag,plat}} = \int_{t_{initial}}^{t_{final}} F_{drag}(t)U(t)dt = F_{drag}U_f T_{plat} \tag{5}$$

The drag force is calculated as:

$$F_{drag} = \frac{1}{2}\rho_w S_{wet} C_D U(t)^2 = \frac{1}{2}\rho_w S_{wet} C_D U_f^2 \tag{6}$$

where $\rho$ is the density of seawater ($\rho$ = 1027 kg/m$^3$); $S_{wet}$ is the surface area of the body that is in contact with the water (*Fish, 1993*; *Fish, 1998*; *Woodward et al., 2006*) calculated as:

$$S_{wet} = 0.08 M_{body}^{0.65} \tag{7}$$

The coefficient of drag ($C_D$) is estimated using an expression inspired by empirical testing of airship aerodynamics (*Blevins, 1984* p. 353; *Fish and Rohr, 1999*; *Gleiss et al., 2015*; *Gleiss et al., 2017*; *Hoerner, 1965* p. 6–17; *Kooyman, 2012* p. 131):

$$C_D = \tilde{F} \underbrace{\left[\frac{0.072}{(R_e)^{0.2}}\right]}_{viscous\ friction} \underbrace{\left[1 + 1.5\left[\frac{w_{max}}{L_{body}}\right]^{1.5} + 7.0\left[\frac{w_{max}}{L_{body}}\right]^{3}\right]}_{pressure\ gradient} \tag{8}$$

which is dependent on velocity ($U$) and accounts for the friction between the body and its boundary layer, and the pressure gradient caused by the near-wake turbulence (*Goldbogen et al., 2015*). The friction adjustment assumes that the whale is moving in a high Reynolds number flow regime ($R_e > 10^6$), and it depends on the Reynolds number:

$$R_e = \frac{L_{body}U(t)}{\nu} = \frac{L_{body}U_f}{\nu} \tag{9}$$

where $\nu$ is the kinematic viscosity of the water. The pressure gradient adjustment depends on the body length and width. Finally, $F$ is an amplification factor used to correct for the extra drag created by the heaving tail and body. Studies of thrust production in dolphins (*Fish, 1993*; *Fish, 1998*) suggest that at $R_e \sim 10^7$, $F$ is between 1 and 3 and therefore we use $F$=2. When swimming horizontally near the surface, CD includes another amplification factor ($\gamma$) to account for wave drag created by the body. However, during most breaching accelerations the body is pitched steeply upwards as the whale swims upwards and therefore no wave drag is created at the surface and $\gamma$ is not included in the equation.

Finally, combining *Equations 5-9* results in the equation for the mechanical work required to overcome drag, when velocity is constant ($W_{drag,\ plat}$):

$$W_{\mathrm{drag,plat}} = \tilde{F}\frac{1}{2}\rho S_{wet}\left[\frac{0.072}{(R_{e\,\mathrm{at\,Uf}})^{0.2}}\right]\left[1 + 1.5\left[\frac{w_{max}}{L_{body}}\right]^{1.5} + 7.0\left[\frac{w_{max}}{L_{body}}\right]^{3}\right]U_f^3 T_{plat} \tag{10}$$

## Drag incurred at constant acceleration

During the acceleration phase, velocity increases with time ($U(t)$, from $U_i$ to $U_f$). The drag force depends on velocity and *Equation 4* cannot be simplified:

$$W_{\mathrm{drag,acc}} = \int_{t_{initial}}^{t_{final}} F_{drag}(t)U(t)dt \tag{11}$$

The drag force is calculated as:

$$F_{drag} = \frac{1}{2}\rho_w S_{wet} C_D U(t)^2 + M_{added}\frac{dU}{dt} \tag{12}$$

where the first term is similar to *Equation 6*. The second term is the acceleration reaction force (*Denny, 1993* p. 43), which accounts for entrained water that must be accelerated with the body. $M_{added}$ is the mass of the entrained water approximated with the following equation:

$$M_{added} = kM_{body} = 0.045M_{body} \tag{13}$$

Where $k$ is the added mass coefficient calculated from inviscid hydrodynamic theory and is approximated as 0.045 for a whale-shaped object (*Gleiss et al., 2017*; *Lamb, 1932* p. 154–155). Combining *Equation 11* with *Equation 12* gives:

$$W_{\text{drag, acc}} = \int_{t_{initial}}^{t_{final}} \frac{1}{2}\rho_w S_{wet} C_D U(t)^2 U(t)\,dt + \int_{t_{initial}}^{t_{final}} M_{added}\frac{dU}{dt}U(t)\,dt \tag{14}$$

integrating the second term results in:

$$\frac{1}{2}M_{added}\left(U_f^2 - U_i^2\right) \tag{15}$$

Assuming that the whale stays in a high Reynolds number flow regime ($R_e > 10^6$) for the entire acceleration, the first term combined with *Equations 8 and 9* becomes:

$$\int_{t_{initial}}^{t_{final}} \frac{1}{2}\rho_w S_{wet} C_D U(t)^2 U(t)\,dt$$
$$= \int_{t_{initial}}^{t_{final}} \tilde{F}\frac{1}{2}\rho S_{wet}\left[0.072\left[\frac{\nu}{L_{body}U(t)}\right]^{0.2}\right]\left[1 + 1.5\left[\frac{w_{max}}{L_{body}}\right]^{1.5} + 7.0\left[\frac{w_{max}}{L_{body}}\right]^3\right]U(t)^3\,dt \tag{16}$$

rearranged this is:

$$\tilde{F}\frac{1}{2}\rho S_{wet}\left[0.072\left[\frac{\nu}{L_{body}}\right]^{0.2}\right]\left[1 + 1.5\left[\frac{w_{max}}{L_{body}}\right]^{1.5} + 7.0\left[\frac{w_{max}}{L_{body}}\right]^3\right]\int_{t_{initial}}^{t_{final}} U(t)^{2.8}\,dt \tag{17}$$

The velocity ($U(t)$) is calculated using the average acceleration ($a_{avg}$):

$$U(t) = U_i + a_{avg}t = U_i + \frac{(U_f - U_i)}{T_{acc}}t \tag{18}$$

The derivative of velocity with respect to time is:

$$\frac{dU}{dt} = \frac{(U_f - U_i)}{T_{acc}} \tag{19}$$

rearranged:

$$dt = dU\frac{T_{acc}}{(U_f - U_i)} \tag{20}$$

which can be substituted into *Equation 18* in order to obtain the integral with respect to velocity:

$$\tilde{F}\frac{1}{2}\rho S_{wet}\left[0.072\left[\frac{\nu}{L_{body}}\right]^{0.2}\right]\left[1 + 1.5\left[\frac{w_{max}}{L_{body}}\right]^{1.5} + 7.0\left[\frac{w_{max}}{L_{body}}\right]^3\right]\int_{U_{initial}}^{U_{final}} U(t)^{2.8}\frac{T_{acc}}{(U_f - U_i)}\,dU \tag{21}$$

evaluating the integral:

$$\tilde{F}\frac{1}{2}\rho S_{wet}\left[0.072\left[\frac{\nu}{L_{body}}\right]^{0.2}\right]\left[1 + 1.5\left[\frac{w_{max}}{L_{body}}\right]^{1.5} + 7.0\left[\frac{w_{max}}{L_{body}}\right]^3\right]\frac{1}{3.8}U(t)^{3.8}\Big|_{U_i}^{U_f}\frac{T_{acc}}{(U_f - U_i)} \tag{22}$$

or:

$$\tilde{F}\frac{1}{2}\rho S_{wet}\left[0.072\left[\frac{\nu}{L_{body}}\right]^{0.2}\right]\left[1+1.5\left[\frac{w_{max}}{L_{body}}\right]^{1.5}+7.0\left[\frac{w_{max}}{L_{body}}\right]^{3}\right]\left[\frac{\left(U_f^{3.8}-U_i^{3.8}\right)}{3.8\left(U_f-U_i\right)}T_{acc}\right] \tag{23}$$

to reintroduce the Reynolds number, multiply by $U_f^{0.2}/U_f^{0.2}$:

$$\tilde{F}\frac{1}{2}\rho S_{wet}\left[0.072\left[\frac{\nu}{L_{body}U_f}\right]^{0.2}\right]\left[1+1.5\left[\frac{w_{max}}{L_{body}}\right]^{1.5}+7.0\left[\frac{w_{max}}{L_{body}}\right]^{3}\right]\left[\frac{\left(U_f^{3.8}-U_i^{3.8}\right)}{3.8\left(U_f-U_i\right)}U_f^{0.2}T_{acc}\right] \tag{24}$$

The equation for the work done against drag during the acceleration phase (*Equation 14*) becomes:

$$W_{\text{drag, acc}} = \tilde{F}\frac{1}{2}\rho S_{wet}\left[\frac{0.072}{\left(R_{\text{eat}Uf}\right)^{0.2}}\right]\left[1+1.5\left[\frac{w_{max}}{L_{body}}\right]^{1.5}+7.0\left[\frac{w_{max}}{L_{body}}\right]^{3}\right]\left[\frac{\left(U_f^{3.8}-U_i^{3.8}\right)}{3.8\left(U_f-U_i\right)}U_f^{0.2}T_{acc}\right]+$$
$$\frac{1}{2}M_{added}\left(U_f^2-U_i^2\right) \tag{25}$$

## Metabolic expenditure

To convert from mechanical energy expenditure to metabolic energy expenditure, the mechanical work done by fluking is multiplied by coefficients to account for energy lost due to metabolic ($\eta_{metab}$ = 0.25) and propulsive ($\eta_{prop}$ = 0.75) efficiency. *Equation 3* becomes:

$$W_{\text{metab, total}} = \frac{1}{\eta_{metab}\eta_{\text{prop}}}\left[\frac{1}{2}M_{body}\left(U_f^2-U_i^2\right)+W_{\text{drag, acc}}+W_{\text{drag, plat}}\right] \tag{26}$$

Combining *Equation 26* with *Equations 10 and 25* yields the final equation for calculating the metabolic work needed for a whale to accelerate from $U_i$ to $U_f$ in time $T_{acc}$, and maintain the final velocity for $T_{plat}$:

$$W_{\text{metab, total}} = \frac{1}{\eta_{metab}\eta_{prop}}\frac{1}{2}M_{body}\left(U_f^2-U_i^2\right)+$$
$$\frac{1}{\eta_{metab}\eta_{prop}}\tilde{F}\frac{1}{2}\rho S_{wet}\left[\frac{0.072}{\left(R_{\text{e at Uf}}\right)^{0.2}}\right]\left[1+1.5\left[\frac{w_{max}}{L_{body}}\right]^{1.5}+7.0\left[\frac{w_{max}}{L_{body}}\right]^{3}\right]\left[\frac{\left(U_f^{3.8}-U_i^{3.8}\right)}{3.8\left(U_f-U_i\right)}U_f^{0.2}T_{acc}\right]+$$
$$\frac{1}{\eta_{metab}\eta_{prop}}\frac{1}{2}M_{added}\left(U_f^2-U_i^2\right)+$$
$$\frac{1}{\eta_{metab}\eta_{prop}}\tilde{F}\frac{1}{2}\rho S_{wet}\left[\frac{0.072}{\left(R_{\text{eat}Uf}\right)^{0.2}}\right]\left[1+1.5\left[\frac{w_{max}}{L_{body}}\right]^{1.5}+7.0\left[\frac{w_{max}}{L_{body}}\right]^{3}\right]U_f^3 T_{plat} \tag{27}$$

## The relative costs of breaching and lunging

The costs of breaching and lunging were compared with estimates of daily field metabolic rate ($FMR_{\text{daily}}$) of humpback whales. The metabolic rates of large whales are poorly understood and therefore we used two separate estimates of $FMR_{\text{daily}}$ that represent possible lower and upper bounds of daily energy usage. The lower bound was calculated using the scaling relationship put forth by *Williams and Maresh (2015)*:

$$FMR_{WM} = 3511 \times m^{0.45} \tag{28}$$

Where $FMR_{\text{daily}}$ is kJ/day and $m$ is mass in kilograms. The upper bound was calculated using the scaling relationship provided by *Nagy (2005)* for terrestrial mammals, multiplied by 1.5 to account for the purported elevated metabolic rate of marine mammals.

$$FMR_{WM} = 1.5 \times 2.25 \times (1000 \times m)^{0.808} \tag{29}$$

Both of these scaling relationships accurately predict the $FMR_{\text{daily}}$ of harbor porpoises (*Rojano-Doñate et al., 2018*), but which one applies to larger cetaceans remains unknown.

For each breach and lunge we present the total energy expended (MJ), the maximum mechanical power output (kW, $W_{metab}\ \eta_{metab}\ T_{acc}^{-1}$, calculated during the last second of the linear acceleration phase), the mass-specific energy expended (J/kg), maximum mass-specific mechanical power output (W/kg), and the energetic cost relative to both calculations of $FMR_{\text{daily}}$ (%). Because of the large

magnitudes involved, we estimated body mass (*Lockyer, 1976*) to the nearest 1000 kg and calculated energy, power, and percentages with a precision of two significant figures. The kinematic and morphological parameters used for the energetic calculations can be found in the supplementary materials (*Supplementary file 1B*).

## Acknowledgements

We thank Megan Jensen for her ideas, efforts in the inception of this paper, and her initial data analysis. We thank the captains and crews of the RV John Martin, the RV Truth, the RV Fluke, the RV Auk, and the Roxy for field logistical support, and the teams from Cascadia Research Collective, Moss Landing Marine Laboratory for field efforts. We also thank the Hawaiian Islands Humpback Whale National Marine Sanctuary, the Stellwagen Bank National Marine Sanctuary, and the Monterey Bay National Marine Sanctuary for their assistance and support of these field efforts. We thank Ed Lyman, David Mattila, Asger Hansen, Bertel Møhl, Franz Hutschenreuter, Rui Prieto, the 'vigias', the whale watching companies from Faial and Pico, Lisa Steiner, and Natacha Aguilar Soto for their field assistance.

## Additional information

### Funding

| Funder | Grant reference number | Author |
| --- | --- | --- |
| National Science Foundation | IOS- 484 1656676 | Jean Potvin<br>John Calambokidis<br>Frank E Fish<br>Ari S Friedlaender<br>Jeremy A Goldbogen |
| National Science Foundation | OPP-1644209 | Ari S Friedlaender<br>Jeremy A Goldbogen |
| Office of Naval Research | N000141612477 | Jeremy A Goldbogen |
| Office of Naval Research | N00014080630 | Susan E Parks |
| University of Hawai'i Sea Grant, SOEST | NA09OAR4170600 | Alison K Stimpert |
| Danish Research Council | | Peter T Madsen |
| Carlsberg Foundation | | Peter T Madsen |
| Volgenau Foundation | | David N Wiley |
| International Fund for Animal Welfare | | David N Wiley |
| National Marine Sanctuary Foundation | | David N Wiley |
| Percy Sladen Memorial Trust | | Jacopo Di Clemente<br>Gwenith S Penry |
| PADI Foundation | | Paolo S Segre<br>Jacopo Di Clemente<br>Gwenith S Penry |
| Society for Marine Mammalogy | | Jacopo Di Clemente<br>Gwenith S Penry |
| Torben og Alice Frimodts Fond | | Jacopo Di Clemente<br>Gwenith S Penry |
| Fundação para a Ciência e a Tecnologia | TRACE-PTDC/MAR/74071/2006 | Cláudia Oliveira |
| Fundação para a Ciência e a Tecnologia | MAPCET-M2.1.2/F/012/2011 | Cláudia Oliveira |

The funders had no role in study design, data collection and interpretation, or the decision to submit the work for publication.

## Author contributions
Paolo S Segre, Conceptualization, Data curation, Software, Formal analysis, Investigation, Methodology; Jean Potvin, Conceptualization, Formal analysis, Funding acquisition, Investigation, Methodology; David E Cade, Data curation, Software, Formal analysis, Investigation; John Calambokidis, Gwenith S Penry, Resources, Funding acquisition, Investigation, Methodology, Project administration; Jacopo Di Clemente, Funding acquisition, Investigation; Frank E Fish, Conceptualization, Formal analysis, Funding acquisition, Methodology; Ari S Friedlaender, Conceptualization, Resources, Funding acquisition, Investigation, Project administration; William T Gough, Shirel R Kahane-Rapport, Data curation, Formal analysis, Investigation; Cláudia Oliveira, Data curation, Formal analysis, Funding acquisition, Investigation, Methodology; Susan E Parks, Resources, Data curation, Formal analysis, Funding acquisition, Investigation; Malene Simon, Resources, Formal analysis, Funding acquisition, Investigation; Alison K Stimpert, Conceptualization, Project Administration, Resources, Data curation, Formal analysis, Funding acquisition, Investigation, Methodology; David N Wiley, Resources, Funding acquisition, Investigation, Project administration; KC Bierlich, Investigation, Methodology; Peter T Madsen, Jeremy A Goldbogen, Conceptualization, Resources, Data curation, Formal analysis, Supervision, Funding acquisition, Investigation, Methodology, Project administration

## Author ORCIDs
Paolo S Segre (iD) https://orcid.org/0000-0002-2396-2670
David E Cade (iD) https://orcid.org/0000-0003-3641-1242
John Calambokidis (iD) https://orcid.org/0000-0002-5028-7172
Jacopo Di Clemente (iD) https://orcid.org/0000-0003-0685-6750
Shirel R Kahane-Rapport (iD) http://orcid.org/0000-0002-5208-1100
Cláudia Oliveira (iD) http://orcid.org/0000-0002-3590-2915
Susan E Parks (iD) https://orcid.org/0000-0001-6663-627X
Alison K Stimpert (iD) https://orcid.org/0000-0002-9400-0305
David N Wiley (iD) https://orcid.org/0000-0001-5490-8645

## Ethics
Animal experimentation: All procedures were conducted under approval of the National Marine Fisheries Service (permits 16111, 19116, 15271, 14809, 14682, 18059); National Marine Sanctuaries (MULTI-2017-007); Marine Mammal Protection Act (775-1875); Department of Environmental Affairs (RES2018/63); Nelson Mandela University animal ethics approval (A18-SCI-ICMR_001); Regional Directorate for Sea Affairs, Autonomous Region of the Azores (49/2010/DRA); and various institutional IACUC committees.

## Decision letter and Author response
Decision letter https://doi.org/10.7554/eLife.51760.sa1
Author response https://doi.org/10.7554/eLife.51760.sa2

## Additional files

### Supplementary files
• Supplementary file 1. Parameters used to calculate the energetics of breaching and lunge feeding, in humpback whales. (A) Lower and upper bounds of daily Field Metabolic Rate ($FMR_{daily}$) for five humpback whales across a range of sizes. $FMR_{daily, WM}$ was calculated using the equation for marine mammal $FMR_{daily}$ proposed by *Williams and Maresh (2015)*. $FMR_{daily, Nagy}$ was calculated using the equation for terrestrial mammal $FMR_{daily}$ proposed by *Nagy (2005)* and multiplied by 1.5. The cost of a high-performance breach and a single high-performance lunge are expressed as percentage of daily energy budget. B) Kinematic and morphological parameters used to calculate the energetics of breaching and lunge feeding.

• Transparent reporting form

## Data availability
The complete dataset is included in the source data file supplement for figure 5. For the energetic model, all of the calculations can be recreated using the numbers provided in Supplementary file 1A and 1B.

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
