## [Decision Letter]

**Acceptance summary:**

Whales are among the largest living species. As such, they perform some of the most energy-expensive maneuvers in nature; like breaching, an iconic animal behavior that tests muscle fibers' physiological limits. Segre et al., put together a nice study, using state-of-the-art sensors and cameras to explore the kinematics and energetics of breaching. The authors found that the energetic cost of breaching is high enough that repeated breaching events may signal body condition. Moreover, they found breaching energetics may be imposing an upper limit on whale body size. The strength of this study lies in the usage of biology, scaling, and engineering principles, to understand the ecology and behavior of these endangered species. This work is thus of interest to physiologists, ecologists, and conservationists.

**Decision letter after peer review:**

Thank you for submitting your article "Energetic and physical limitations on the breaching performance of large whales" for consideration by *eLife*. Your article has been reviewed by three peer reviewers, and the evaluation has been overseen by a Reviewing Editor and Detlef Weigel as the Senior Editor. The following individuals involved in review of your submission have agreed to reveal their identity: Alexander Werth (Reviewer 2).

The reviewers have discussed the reviews with one another and the Reviewing Editor has drafted this decision to help you prepare a revised submission.

Summary:

The authors put together a nice study exploring the many ways some whale species breach. They brought the insightful idea of using tags to study the kinematics and energetics of breaching. As, the authors write, "most of what we know about breaching comes from above-water… observations", tag data can fill some major gaps in our understanding of this extreme, and extremely interesting, behavior. The current study includes new and important information, such as the finding that sperm whales do not need long ascents before breaching.

One of the reviewers especially enjoyed reading about effects of body mass vs. body length on whales' breaching abilities, as well as about exit angle, rolls and backflips, and so on. The comparisons of breaching's energy expenditure/power output to draft horses and human marathon runners might put off some reviewers, but it is comparison, which truly helps to demonstrate the huge amounts of energy involved here. Likewise the analogy of projectile motion to the body length vs. mass issue is also useful.

The upcoming comments are drawn directly from reviewers' answers. In general, most comments denote the general perception that the manuscript is well-framed and the data is as great as it gets. But, the conclusions need to be more succinct. This may require the authors to re-state their case and probably re-define the central message of the paper. Then I expect the authors to see the proposed revision as an opportunity to accomplish a nice text that has broader appeal, but at the same time moves the field forward.

Essential revisions:

1) My field observations of whales are skewed much more toward higher-latitude feeding grounds than breeding/calving grounds, but I am much more familiar with breaches (both from my own personal sightings as well as records reported by others) at feeding grounds.

The reason I raise this point is that there is a focus here on breeding/calving grounds as if the social signaling is especially important in breeding grounds. However, as noted above, I have quite frequently seen whales breach far from their low-latitude winter breeding grounds. Also, the tagging locations listed at the start of the Materials and methods section makes clear that many of the tags were deployed in locations (e.g., Antarctica, Cape Cod) that are not breeding grounds.

I also realize this is somewhat species-specific. For example, I don't think I've ever heard of gray whales breaching in feeding areas, though know gray whales breach in breeding grounds. In contrast, breaching in feeding grounds is quite common for humpback whales.

I'm not necessarily disputing the authors' contention that breaching is an important signaling phenomenon that is important in breeding groups, but I worry a bit that readers less familiar with field observations of whales might conclude that breaching occurs primarily (or solely) on breeding grounds. I am sure the authors don't believe this, and I'm not saying they mean to say this, but I think a less-educated reader might unfortunately conclude this. I urge the authors to think about this point and address it in some way.

2) I have heard nearly a dozen different explanations for breaching (not all of them exclusive), and I have to say I really like the honest signaling hypothesis. However, I wonder if it would be useful for the authors to list and explain, in a bit more (but not extensive) detail early on in this manuscript, some of the other possibilities that have attracted scrutiny.

Specifically, I wonder if the authors might comment on the possibility that breaching might sometimes be related to parasites, either as a behavioral means to detach ectoparasites or as a result of endoparasites in whale ears or sinuses, etc. I think the authors should not ignore this idea because they specifically note that a "breach was likely a response to tagging, since it occurred immediately after the deployment." Might this not serve, even circumstantially, as evidence for a parasite-induced cause of breaching, and more specifically for breaching as a means to remove troublesome or annoying items on the body's external surface?

3) With regard to the honest signaling hypothesis-and also relating to the effects of body length/size on breaching ability-I find it interesting that in large cetaceans where there is notable sexual dimorphism (sperm whales, for example), the smaller whales are more commonly observed breaching. The Introduction explains that female sperm whales "regularly breach" whereas larger males "breach very infrequently." Social signaling theory is a vast, robust area of evolutionary biology, but my (admittedly limited) understanding is that such signaling, although not always related to sexual selection, is nonetheless predominantly or generally performed by males. I wonder if the authors might comment on how sperm whales, and other cetacean species where females "signal" (=breach) more commonly, relate to this, and whether it is at all significant…

4) I don't doubt the authors' conclusion that breaching ability relates to (shorter) body length, which I think makes sense (and for which they make a strong argument), but at the same time I can't help thinking that humpbacks breach more than other balaenopterids-even smaller rorquals such as minke whales-because humpbacks are simply much more acrobatic than other balaenopterids, and thus this breaching extends humpbacks' general underwater pattern of remarkable mobility and maneuverability above the air-water interface as well. Over 80% of your breach events (152/187) came from humpbacks, including 57% of the breaches from three young humpbacks, but I think this is not due to something rare or special about those three whales-all humpbacks are known for being highly mobile under, at, and above the water surface.

a) In this regard, do the authors care to comment on whether breaching is common in humpbacks due to their body size/shape/length, etc., or perhaps to behavioral, social, or other factors (not that these are exclusive)…?

b) Second, I wonder if a comparison just within this species (i.e., of the 28 tagged humpbacks) might be particularly useful in elucidating the influence of size/length, and whether this merits special mention and/or a figure or figure panel? And yes, I understand that Figure 6 directly shows the influence of body mass in humpbacks.

5) Back to the idea of length relating to breaching ability, I've never even heard of young blue or fin whales breaching, which leads me to think this might be species-specific as well as length-specific. Do the authors know anything (even from previous studies by other scientists) about whether shorter, younger individuals of the biggest whale species occasionally breach?

6) This also leads me to wonder about age-related effects. Do younger whales (e.g., humpbacks) breach more frequently than older, longer/heavier ones?

7) There is good description of the results with regard to exit angle, pitch, roll, and so on. I wonder if the authors found any relationship between body size (mass)/length and whether a breach involves a backflip, etc.?

8) With regard to rotation during breaching, there appears to be no correlation whatsoever between exit speed and roll angle (Figure 5D). Is there any correlation between exit pitch and roll angle? I know rolling has been studied extensively in spinner dolphins (e.g., Fish et al., 2006).

9) The manuscript's treatment of potential biomechanical limits to breaching is, in my view, well done in terms of methodology, analysis, and conclusions. I wonder how long it takes a whale to "recover" after breaching. I have seen a humpback whale breach twice in rapid succession (I am sure it was the same whale). Does the tag data analyzed for this study tell anything about how much (or how little) time whales take between multiple breaches? I expect the data from the three very active juvenile humpbacks that breached 106 times might reveal something about this behavior's relation to limits from muscle vs. FMR.

10) I like to glance at the figures after reading the Abstract, and Figure 6 wasn't immediately understandable to me based on the information contained in the panels and legend. Specifically, I wondered what the% was referring to. I thought the figure was showing information about variation in breaching costs and speed with increasing body size in a general sense (e.g., from aggregated or average data), not displaying data from five specific breaches, as I quickly realized from Table 2. The text doesn't help a lot. I suggest that the Figure 6 legend be tweaked to make the% emergence idea clearer.

11) Using the authors own words, this study addresses the following questions:

a) What are the underwater trajectories and fluking patterns that different species of large whales use to perform breaches?

b) What are the energetic costs of breaching, and how do they scale with body size?

c) Do energetic or physical constraints impose fundamental limits on the breaching behaviors of large whales?

The questions are stated in a logical order as the answer to the first one is needed to answer the second, which, in turn, needed to answer the last. Nonetheless, I am not satisfied with the answers to the last two questions – I would have expected a sharper message from an *eLife* paper.

12) The writing style is far from being succinct. For example, there is a long verbal description of breaching trajectories. Given the figure on which these trajectories are displayed, I do not need this description. I am afraid that vital information can be easily lost between too many words.

13).…Consequently, I am not ruling out that I missed it, but I could not find neither a definition nor a trajectory of a feeding event. What is the difference between a feeding event and a partial breach? Because a feeding event can be an excellent unit of energy – at least for humpback whales – it has to be defined and the respective trajectories shown for comparison.

14) Daily energy expenditure is a bad choice for the unit of energy because it remained practically undefined. The two estimates suggested in the paper differ (for a 10-ton whale) by more than an order of magnitude! I suggest leaving it out of the paper. An energy expenditure in a single feeding event (I am talking about mechanical energy only) is a much better unit. An average amount of energy in a single gulp is a good unit as well.

15) Why gulping the water during a feeding event is excluded from its energy expenditure? For me it is an inseparable part of the event.

16) What is a definition of “breach”? The answer here will affect the conclusion of this study. Length is a hindrance only if the “breach” is defined as rising a certain portion of body length out of water. Is it?

17) Energy expenditure calculation could have been done better, and written better – I apologize.

First, define the drag. It is commonly written as D=(1/2)ρ v^2 S CD, where S is a certain reference area and CD is the associated drag coefficient (note that it greatly increases when the mouth opens). CD is a weak function of speed. Your equation (8) sets it. There is no need to write it explicitly thereafter. Next, write the Newton second law. For a neutrally buoyant body it is: ma*(dv/dt)=T-D, where ma is the apparent mass, the sum of the real mass and the added one. The added mass is negligible as compared with the real mass, and considering estimation uncertainties, can be safely ignored. Rewriting this equation as T=D+ma*dv/dt, and integrating it along the swimming path, yields \int{T*v,t=0 to t=end} = \int{D*v, t=0 to t=end} +ma*(v(end)^2-v(0)^2). The expression on the left is the mechanical energy spent; the integral on the right can be evaluated numerically (D is a function of v only) without any additional assumptions. After all, the speed is known at every instant. “trapz” function in Matlab will do the job. The write-up in the paper needs not be longer than this comment. It is counterproductive to explicate constants in an equation, especially if they are empirical.

18) I do not believe in doubling the drag coefficient during swimming. As mentioned in the paper itself, drag can be divided into viscous (friction) drag and pressure drag. The former is practically independent on the shape of the body, whereas the latter can increase only if body flex induces flow separation. There is no evidence that flow separation occurs over swimming fishes, and therefore an increase in drag is unjustified. Liu, Barazani, Triantafyllou are just a few gentlemen that were working on this in the last 15 years. I suggest revising.

19) Speed measurement is in the heart of this paper, but very little attention is given in the paper to its calibration procedure. I am skeptical about using an acceleration signal for speed measurement after it has been down-sampled to 25 (and possibly less) HZ. The method was designed with turbulence noise in mind, and this frequency seems too low to be effectively associated with it Adding a supplementary on speed calibration may help.

20) I am not sure that the conclusion that mass specific energetic cost increases with size (and hence large whales do not breach) was substantiated,. Figure 6 is hardly convincing. I suggest revising.

---

## [Author Response]

Summary:The authors put together a nice study exploring the many ways some whale species breach. They brought the insightful idea of using tags to study the kinematics and energetics of breaching. As, the authors write, "most of what we know about breaching comes from above-water… observations", tag data can fill some major gaps in our understanding of this extreme, and extremely interesting, behavior. The current study includes new and important information, such as the finding that sperm whales do not need long ascents before breaching.One of the reviewers especially enjoyed reading about effects of body mass vs. body length on whales' breaching abilities, as well as about exit angle, rolls and backflips, and so on. The comparisons of breaching's energy expenditure/power output to draft horses and human marathon runners might put off some reviewers, but it is comparison, which truly helps to demonstrate the huge amounts of energy involved here. Likewise the analogy of projectile motion to the body length vs. mass issue is also useful.The upcoming comments are drawn directly from reviewers' answers. In general, most comments denote the general perception that the manuscript is well-framed and the data is as great as it gets. But, the conclusions need to be more succinct. This may require the authors to re-state their case and probably re-define the central message of the paper. Then I expect the authors to see the proposed revision as an opportunity to accomplish a nice text that has broader appeal, but at the same time moves the field forward.

We thank the reviewers and the editor for the positive comments, and we agree that the conclusions could be clarified to better express the central message of the manuscript. We have substantially revised the Discussion to directly answer the three questions that we propose in the Introduction and that we restate in the first paragraph of the Discussion. As the reviewer states in comment 11, these questions follow a logical order and must be answered sequentially to guide the reader through our thought process to the final conclusions of the paper. We have also added a new panel to Figure 6 which presents the maximum mass-specific power and now allows the reader to sequentially follow along with the conclusions to question 3. The first two paragraphs of the subheading entitled Does body size limit breaching performance?section now read:

"On a mass-specific basis, the cost of breaching also increases with body size (Figure 6A) and this increase is largely driven by the higher speeds required to emerge from the water (Figure 6B). […] It is not clear whether blue whales can even reach this speed (Gough et al., 2019), which may be limited by both muscular power output and the hydrodynamic limits of lunate tail propulsion (Iosilevskii and Weihs, 2008)."

The last paragraph of the Discussion now reads:

"In conclusion, our results suggest an underlying biomechanical explanation for the factors that limit intra-specific and inter-specific breaching ability in large whales. […] The confluence of muscle contractile properties, hydrodynamic limitations of lunate tail propulsion, and the higher speeds required for longer whales to emerge from the water likely imposes an upper limit to the body size and effectiveness of breaching whales."

Essential revisions:1) My field observations of whales are skewed much more toward higher-latitude feeding grounds than breeding/calving grounds, but I am much more familiar with breaches (both from my own personal sightings as well as records reported by others) at feeding grounds.The reason I raise this point is that there is a focus here on breeding/calving grounds as if the social signaling is especially important in breeding grounds. However, as noted above, I have quite frequently seen whales breach far from their low-latitude winter breeding grounds. Also, the tagging locations listed at the start of the Materials and methods section makes clear that many of the tags were deployed in locations (e.g., Antarctica, Cape Cod) that are not breeding grounds.I also realize this is somewhat species-specific. For example, I don't think I've ever heard of gray whales breaching in feeding areas, though know gray whales breach in breeding grounds. In contrast, breaching in feeding grounds is quite common for humpback whales.I'm not necessarily disputing the authors' contention that breaching is an important signaling phenomenon that is important in breeding groups, but I worry a bit that readers less familiar with field observations of whales might conclude that breaching occurs primarily (or solely) on breeding grounds. I am sure the authors don't believe this, and I'm not saying they mean to say this, but I think a less-educated reader might unfortunately conclude this. I urge the authors to think about this point and address it in some way.

We appreciate the reviewer's concern and have made the following changes. The Introduction has been changed to *"*since species with complex social structures breach frequently". The original intent of this sentence was to convey the idea that species with distinct breeding grounds have complex social structures, but this is a more direct way to say it. We have also removed the reference to breeding grounds, where it does not add to the message of the paragraph. We did not change subsection “Is breaching energetically expensive?” because the focus of this section is that capital breeders that breach regularly at their breeding grounds cannot easily recoup the energy expended. Also, to answer the reviewer's question, we have recorded several instances of gray whales breaching in the Puget Sound region (although only one animal was tagged).

2) I have heard nearly a dozen different explanations for breaching (not all of them exclusive), and I have to say I really like the honest signaling hypothesis. However, I wonder if it would be useful for the authors to list and explain, in a bit more (but not extensive) detail early on in this manuscript, some of the other possibilities that have attracted scrutiny.Specifically, I wonder if the authors might comment on the possibility that breaching might sometimes be related to parasites, either as a behavioral means to detach ectoparasites or as a result of endoparasites in whale ears or sinuses, etc. I think the authors should not ignore this idea because they specifically note that a "breach was likely a response to tagging, since it occurred immediately after the deployment." Might this not serve, even circumstantially, as evidence for a parasite-induced cause of breaching, and more specifically for breaching as a means to remove troublesome or annoying items on the body's external surface?

Good point: to provide some more background information on the hypothesized purposes of breaching we have added the following sentence:

"The reasons why large whales breach remain unclear, with possible, non-exclusive explanations ranging from ectoparasite removal (as seen in dolphins, Fish et al., 2006) to play (juvenile whales breach frequently, Würsig et al., 1989). Another, commonly held explanation is that in large whales, aerial displays are a form of social communication."

We agree that this information is important, but are reluctant to explain the breach recorded from the gray whale as an attempt to remove the tag. The whale only breached once and then returned to a calm state, sitting near the bottom of the sound. The deployment lasted for 17 hours and the whale made no further breaches or attempts to dislodge the tag. For this reason, we think that a more likely explanation for the breach was as a signal of displeasure.

3) With regard to the honest signaling hypothesis-and also relating to the effects of body length/size on breaching ability-I find it interesting that in large cetaceans where there is notable sexual dimorphism (sperm whales, for example), the smaller whales are more commonly observed breaching. The Introduction explains that female sperm whales "regularly breach" whereas larger males "breach very infrequently." Social signaling theory is a vast, robust area of evolutionary biology, but my (admittedly limited) understanding is that such signaling, although not always related to sexual selection, is nonetheless predominantly or generally performed by males. I wonder if the authors might comment on how sperm whales, and other cetacean species where females "signal" (=breach) more commonly, relate to this, and whether it is at all significant…

To answer the reviewer's question: male sperm whales use unique clicks that likely convey their size to other individuals. Please see the next section regarding the use of breaching for signaling.

4) I don't doubt the authors' conclusion that breaching ability relates to (shorter) body length, which I think makes sense (and for which they make a strong argument), but at the same time I can't help thinking that humpbacks breach more than other balaenopterids-even smaller rorquals such as minke whales-because humpbacks are simply much more acrobatic than other balaenopterids, and thus this breaching extends humpbacks' general underwater pattern of remarkable mobility and maneuverability above the air-water interface as well. Over 80% of your breach events (152/187) came from humpbacks, including 57% of the breaches from three young humpbacks, but I think this is not due to something rare or special about those three whales-all humpbacks are known for being highly mobile under, at, and above the water surface.

We agree with the reviewer that the reason why certain species breach frequently is more complex than a simple physical argument. Rather, it is more likely a combination of complex social structures and the abilities to breach that predispose certain species to incorporate breaching as a method of signaling. Species-level maneuverability also likely plays an important role, however, the comparative maneuvering abilities of large whales are currently poorly understood and remain somewhat anecdotal. To clarify these issues we have changed the penultimate paragraph to:

"The physical and behavioral limitations on breaching performance are likely more complex and nuanced than the first approximations presented here. […] Meanwhile, the physical ability to breach efficiently combined with a complex social structure and high levels of innate maneuverability may have predisposed certain species, such as humpback whales, to incorporate breaching as a form of communication."

a) In this regard, do the authors care to comment on whether breaching is common in humpbacks due to their body size/shape/length, etc., or perhaps to behavioral, social, or other factors (not that these are exclusive)…?

Good question, please see above response.

b) Second, I wonder if a comparison just within this species (i.e., of the 28 tagged humpbacks) might be particularly useful in elucidating the influence of size/length, and whether this merits special mention and/or a figure or figure panel? And yes, I understand that Figure 6 directly shows the influence of body mass in humpbacks.

This was our original intent, however we only had body length measurements for a small subset of the 28 tagged humpback whales. Figure 6 includes all of the whales with known body lengths and high-performance breaches in our dataset.

5) Back to the idea of length relating to breaching ability, I've never even heard of young blue or fin whales breaching, which leads me to think this might be species-specific as well as length-specific. Do the authors know anything (even from previous studies by other scientists) about whether shorter, younger individuals of the biggest whale species occasionally breach?

There is some very sparse, anecdotal evidence that blue whale and fin whale juveniles can breach, albeit very rarely. Whitehead, 1986, lists the propensity of large rorquals to breach as follows: blue – almost never; sei – almost never; finback – rare. One of our co-authors (J.C.) known for his experience around whales, has not seen juvenile blue or fin whales breaching (except for the racing blue whales described in the manuscript), although given the location of his work he does not often encounter juvenile blue and fin whales. Personally, I (P.S.S) was on a boat in a newly discovered breeding ground for blue whales when a juvenile blue whale breached, off in the distance. The entire crew was surprised by the event, and there was much discussion on whether that was actually a juvenile blue whale or something else. All this is to say that if juvenile blue, fin, and sei whales breach, it is a very rare event, and we agree with the reviewer that there is likely a combination of physical and species-level behavioral limitations on breaching. We hope that we adequately addressed this in comment 4 and the associated changes to the manuscript.

6) This also leads me to wonder about age-related effects. Do younger whales (e.g., humpbacks) breach more frequently than older, longer/heavier ones?

Yes, this is well documented (by Whitehead, Wursig, and others) and is one of the reasons for the hypothesis that breaching is a form of play, for juveniles. We have addressed this in the same sentence that we addressed comment 2.

7) There is good description of the results with regard to exit angle, pitch, roll, and so on. I wonder if the authors found any relationship between body size (mass)/length and whether a breach involves a backflip, etc.?

The sample size of individuals and breaching events performed by whales of known dimensions was too low to be conclusive. Anecdotally from the videos, it seems as if juveniles add more long-axis rotation to their breaches which allows them to emerge from the water in a larger amount of configurations. Meanwhile larger whales spin less often and smaller amounts and so when they emerge upside-down this is a direct result of the 'backflip'. This is discussed briefly in the Discussion.

8) With regard to rotation during breaching, there appears to be no correlation whatsoever between exit speed and roll angle (Figure 5D). Is there any correlation between exit pitch and roll angle? I know rolling has been studied extensively in spinner dolphins (e.g., Fish et al., 2006).

We looked at this and there was no correlation between exit pitch and roll angle.

9) The manuscript's treatment of potential biomechanical limits to breaching is, in my view, well done in terms of methodology, analysis, and conclusions. I wonder how long it takes a whale to "recover" after breaching. I have seen a humpback whale breach twice in rapid succession (I am sure it was the same whale). Does the tag data analyzed for this study tell anything about how much (or how little) time whales take between multiple breaches? I expect the data from the three very active juvenile humpbacks that breached 106 times might reveal something about this behavior's relation to limits from muscle vs. FMR.

The topic of breaching sequences has been studied extensively using traditional focal follow techniques (including much of work from Whitehead and Wursig). For one of the juveniles that we tagged, the shortest time between consecutive breaches was 6.5 seconds and for the other it was 10.5 seconds. The third juvenile breached intermittently. This is certainly a very interesting topic, with the caveat that juveniles also performed many other acrobatic maneuvers in-between breaches and the breaches varied widely in emergence% . The scaling of recovery time with body mass is a worthwhile topic and would probably have some interesting implications, but we believe our dataset is too sparse to accurately pursue this (due to the low number of whales with known body lengths, and low number multiple-breach sequences that could assure us of an accurate minimum time between events). We added the following line as documentation of the time between consecutive breaches:

"For one of the juvenile whales, the shortest time between consecutive breaches was 6.5 seconds."

10) I like to glance at the figures after reading the Abstract, and Figure 6 wasn't immediately understandable to me based on the information contained in the panels and legend. Specifically, I wondered what the% was referring to. I thought the figure was showing information about variation in breaching costs and speed with increasing body size in a general sense (e.g., from aggregated or average data), not displaying data from five specific breaches, as I quickly realized from Table 2. The text doesn't help a lot. I suggest that the Figure 6 legend be tweaked to make the% emergence idea clearer.

We have modified Figure 6 in response to comment 20 and updated the caption to:

"The cost of breaching increases with body size, in humpback whales. A) The mass-specific energy expenditure required to perform high-emergence breaches (blue) and high-performance lunges (red) is shown for five humpback whales of different sizes. […] Both the model and the data show that the mass-specific cost of breaching increases with body size. B) This pattern is largely driven by the higher speeds that larger whales need to emerge from the water. C) To attain the higher speeds required to emerge from the water, larger whales need to generate higher mass-specific power outputs or extend the duration of their trajectories (green numbers)."

11) Using the authors own words, this study addresses the following questions:a) What are the underwater trajectories and fluking patterns that different species of large whales use to perform breaches?b) What are the energetic costs of breaching, and how do they scale with body size?c) Do energetic or physical constraints impose fundamental limits on the breaching behaviors of large whales?The questions are stated in a logical order as the answer to the first one is needed to answer the second, which, in turn, needed to answer the last. Nonetheless, I am not satisfied with the answers to the last two questions – I would have expected a sharper message from an eLife paper.

We agree that the manuscript will be greatly improved by streamlined conclusions. Please see the response to the editor's notes for the detailed changes that we have made. We have substantially revised the concluding paragraph to directly answer the three questions that we propose, in a sequential manner designed to guide the reader to the central message of the paper. As stated in comment 12, we have also moved a significant amount of the descriptive text in the results to a new table.

12) The writing style is far from being succinct. For example, there is a long verbal description of breaching trajectories. Given the figure on which these trajectories are displayed, I do not need this description. I am afraid that vital information can be easily lost between too many words.

We agree, and have converted that paragraph to a new table (Table 2) in order to reduce the length of the text.

13) Consequently, I am not ruling out that I missed it, but I could not find neither a definition nor a trajectory of a feeding event. What is the difference between a feeding event and a partial breach? Because a feeding event can be an excellent unit of energy – at least for humpback whales – it has to be defined and the respective trajectories shown for comparison.

This is a good point, and so we have added the trajectory of a feeding lunge to Figure 3, and added the following text:

"Rorqual whales feed by rapidly accelerating, opening their mouths, and engulfing large volumes of prey-laden water. Although the trajectories used for feeding lunges are highly variable (Cade et al., 2016), lunges are common behaviors that require a rapid acceleration similar to that used for breaching."

14) Daily energy expenditure is a bad choice for the unit of energy because it remained practically undefined. The two estimates suggested in the paper differ (for a 10-ton whale) by more than an order of magnitude! I suggest leaving it out of the paper. An energy expenditure in a single feeding event (I am talking about mechanical energy only) is a much better unit. An average amount of energy in a single gulp is a good unit as well.

The Field Metabolic Rate of large cetaceans has been difficult to quantify and remains somewhat controversial. As explained in the text, there are two main theories for how FMR scales with extreme body mass. We agree that the two competing theories provide very different estimates, however, our results show that under both scaling regimes, the cost of breaching increases disproportionately with body mass. We believe this result is important enough to warrant the paragraph that we devote to this topic (also see Supplemental File 1A).

We agree that the mechanical cost of lunging makes a good comparison for the cost of breaching. The mechanical cost of lunging can either refer to the pre-engulfment acceleration, or the pre-engulfment acceleration and the post-engulfment deceleration (which includes acceleration of engulfed water). If the goal is to compare the energetics cost of two events, the latter is appropriate. If the goal is to compare the energetic costs of two mechanically similar trajectories, then the former is appropriate. For this reason, we do compare the cost of breaching with the cost of the mechanically similar pre-engulfment acceleration phase of lunging (Figure 6 and the old Table 2). This comparison did turn out to be very interesting because, while feeding lunges are generally considered to use 'high-performance' accelerating maneuvers, our results show that even high-speed lunges are relatively cheap compared to breaches (and most lunges feature much slower speeds than the ones used for our comparison).

While we do agree that the energy contained in a single gulp would make a good alternative comparison for the cost of breaching, it is also subject to many uncertainties (high variability in buccal cavity inflation, prey density, prey type, escape response). Meanwhile, its ecological relevance is not as straightforward as FMR. For example, saying that a breach costs X gulps may not be informative for a reader who does not know that a whale may perform 0 – 700+ feeding lunges in a day. We respectfully argue that saying that a breach costs X% of the whale's daily energy budget is a simpler comparison.

15) Why gulping the water during a feeding event is excluded from its energy expenditure? For me it is an inseparable part of the event.

As described in the response to 14, this would represent a different way to compare energetic expenditure (vs mechanical cost of lunging accelerations; energy contained in a gulp; or daily FMR). We chose to focus on the mechanical cost of lunging (pre-engulfment phase) because this represents an accelerating trajectory similar to the accelerations used for breaching.

16) What is a definition of “breach”? The answer here will affect the conclusion of this study. Length is a hindrance only if the “breach” is defined as rising a certain portion of body length out of water. Is it?

We defined breaches per the traditionally used definition (Whitehead, 1985):

"We classified aerial behaviors as full breaches when >40% of the whale emerged from the water (Whitehead, 1985a). The remaining behaviors were classified as partial breaches."

In the original version of this manuscript, the definition was presented earlier. Because we moved the Materials and methods section to the end of the manuscript to fit *eLife*'s formatting requirements, we have added the following clarification to the Results section:

"125 of the breaches were classified as 'full breaches', where >40% of the whale emerged from the water (Whitehead, 1985b); 52 of the breaches were classified as 'partial breaches' (<40% emergence); and 10 were undetermined."

17) Energy expenditure calculation could have been done better, and written better – I apologize.First, define the drag. It is commonly written as D=(1/2)ρ v^2 S CD, where S is a certain reference area and CD is the associated drag coefficient (note that it greatly increases when the mouth opens). CD is a weak function of speed. Your equation (8) sets it. There is no need to write it explicitly thereafter. Next, write the Newton second law. For a neutrally buoyant body it is: ma*(dv/dt)=T-D, where ma is the apparent mass, the sum of the real mass and the added one. The added mass is negligible as compared with the real mass, and considering estimation uncertainties, can be safely ignored. Rewriting this equation as T=D+ma*dv/dt, and integrating it along the swimming path, yields \int{T*v,t=0 to t=end} = \int{D*v, t=0 to t=end} +ma*(v(end)^2-v(0)^2). The expression on the left is the mechanical energy spent; the integral on the right can be evaluated numerically (D is a function of v only) without any additional assumptions. After all, the speed is known at every instant. “trapz” function in Matlab will do the job. The write-up in the paper needs not be longer than this comment. It is counterproductive to explicate constants in an equation, especially if they are empirical.

We appreciate the reviewer's suggestions for simplifying the explanation of the equations in the manuscript. The alternate method the reviewer describes is similar but not equivalent to the method that we use, since it relies on integrating the speed of the entire breaching trajectory. In contrast, our method uses the starting and ending velocities and requires deciding whether the breach follows a linear acceleration or a linear acceleration with a plateau. Both methods result in similar results, although the numbers are not exactly the same. We did try the reviewer's suggestion but upon further consideration, our method allows the reader to use the values from Table S1 to recreate our results. Our method also allows for simple, theoretical trajectories to be constructed (see blue line in revised Figure 6A-C, the maximum power calculations for the new panel Figure 6C, and the analysis of theoretical blue whale breaching velocities). For these reasons we would like to keep our analysis in its current form.

Additionally, although our derivation is lengthy, we believe that providing it is important for allowing the readers to evaluate the final form of the equation. Originally, the derivation was located in the supplementary section, but we moved it to the main text to conform with *eLife*'s format. We also agree that providing the coefficient of drag as a constant would be simpler, however, in our equations Cd is dependent on the Reynolds number and thus, the velocity. Therefore, to perform the integration, Cd must be expanded. After equation 27, Cd_final_ can be substituted back in, but that would require including an additional equation. We would be happy to move the derivation to a Mathematical Model section after the Materials and methods section, if the *eLife* format permits.

18) I do not believe in doubling the drag coefficient during swimming. As mentioned in the paper itself, drag can be divided into viscous (friction) drag and pressure drag. The former is practically independent on the shape of the body, whereas the latter can increase only if body flex induces flow separation. There is no evidence that flow separation occurs over swimming fishes, and therefore an increase in drag is unjustified. Liu, Barazani, Triantafyllou are just a few gentlemen that were working on this in the last 15 years. I suggest revising.

The doubling of the drag coefficient (*F = 2*) does not come from flow separation at the end of the tail (which we agree is very small anyway), but from the flow accelerations imparted by the heaving body. Such doubling was based on the work by Frank Fish on the hydrodynamics of fluking odontocetes (Fish, 1993, 1998). Here Fish used kinematic measurements to calculate the value of the fluking thrust based on the lunate tail thrust-efficiency modeling of Chopra and Kambe, 1977, and Yates, 1983). The results yielded drag coefficients that ended up at 2 to 3 times higher than the drag estimated for same-area flat plates in longitudinal low, a finding that turned out in agreement with similar drag and thrust studies of fish (Blake, 1983, pp. 98-102; see also the short review by Schultz and Webb, 2002).

Chopra, M.G. and Kambe, T. 1977. Hydrodynamics of lunate-tail swimming propulsion. Part 2. J. Fluid Mech. 79, 49–69.

Yates, G.T. 1983. Hydrodynamics of body and caudal fin propulsion. In Fish Biomechanics (ed.P. W. Webb and D. Weihs), pp. 177–213. New York: Praeger.

Schultz, William W., and Paul W. Webb. 2002 Power requirements of swimming: Do new methods resolve old questions?. Integrative and Comparative Biology.

19) Speed measurement is in the heart of this paper, but very little attention is given in the paper to its calibration procedure. I am skeptical about using an acceleration signal for speed measurement after it has been down-sampled to 25 (and possibly less) HZ. The method was designed with turbulence noise in mind, and this frequency seems too low to be effectively associated with it Adding a supplementary on speed calibration may help.

For most of the deployments, speed was calculated using the Orientation Corrected Depth Rate (OCDR). For the humpback breaches used for the scaling analysis we used the accelerometer vibration method performed on the full-resolution, 400Hz data. We apologize for not clarifying this previously, and have made the following change to the text L398:

"For the CATS tag deployments we calibrated the measurements of the background, high frequency accelerometer vibrations (sampled from the 400Hz data) with the orientation-corrected depth rate."

The method is exactly as described in Cade et al., 2018 and numerous other papers that have subsequently used this procedure.

20) I am not sure that the conclusion that mass specific energetic cost increases with size (and hence large whales do not breach) was substantiated,. Figure 6 is hardly convincing. I suggest revising.

We revised Figure 6 to include the speed and cost of idealized breaches performed with uniform trajectories across the range of humpback body sizes. This demonstrates how the data, drawn from real trajectories with differing parameters, compares with the predicted energetics for stereotyped breaches that all achieve 80% emergence.